# Linker histone H1 represses H3 tail acetylation induced by H4 tail acetylation and alters its dynamics
Ayako Furukawa[1,2], Kenta Echigoya[3], Samuel Blazquez [4,5], Masatoshi Wakamori [6], Hideaki Ohtomo [1], Yasuo Tsunaka [1], Takashi Umehara [6,7], Tsuyoshi Terakawa [4], Yoshimasa Takizawa [3,8], Hitoshi Kurumizaka [3,9,10] & Yoshifumi Nishimura [1]✉

The nucleosome is the fundamental chromatin unit, containing two copies of histones H2A, H2B, H3, and H4 wrapped by ~ 146 bp of core DNA plus linker DNA; addition of linker histone H1 forms a chromatosome. Tetra-acetylation of the H4 N-terminal tail (H4-4Kac) enhances H3 N-tail acetylation by altering their mutual dynamics, but how H1 influences these dynamics remains unclear. Using cryo-electron microscopy and coarse-grained molecular dynamics simulations, we show that H4-4Kac and unmodified chromatosomes share essentially identical core histone–DNA structures and similar H3 N-tail dynamics. However, nuclear magnetic resonance spectroscopy reveals that in the H4-4Kac chromatosome, the H3 N-tail adopts a dynamically robust DNA-contact state distinct from that in the unmodified chromatosome, resulting in markedly reduced H3 N-tail acetylation. These findings suggest that linker histone H1 suppresses the progression of euchromatin formation.

The nucleosome is a fundamental repeating unit of chromatin, consisting of a histone octamer composed of two (H2A-H2B) heterodimers and one (H3-H4)₂ tetramer, around which ~146 base pairs (bp) of DNA is wrapped[1], along with several lengths of linker DNA extending of both ends[2]. Additionally, a linker histone H1 binds to the linker-DNA, forming a chromatosome. Chromatin structures are roughly classified into two types, transcription active euchromatin and inactive heterochromatin, depending on the post-translational modifications (PTMs) present on histones. Most PTMs occur in the intrinsically disordered N-tails of histones[3]. For example, the N-tails of histones H3 and H4 in the nucleosome are highly acetylated in euchromatin, but are less acetylated in heterochromatin. Atomic detailed structures of nucleosomes and chromatosomes have been determined by X-ray crystallography[1,4] and cryo-electron microscopy (cryo-EM) methods[5,6]. In most cases, however, the structures of the histone tails remain unresolved, although their dynamics and PTMs have been studied by nuclear magnetic resonance (NMR) spectroscopy[7–21].

Notably, the two-dimensional ¹H and ¹⁵N NMR signals of the H3 N-tail in the nucleosome were closer to those of an H3 N-tail peptide (residues 1–33) fused to the globular protein GB1 in its DNA-bound form than to those of the peptide in its DNA-free form[9], indicating that the H3 N-tail in the nucleosome transiently binds to linker DNA. This conclusion was further supported by H3 N-tail modification experiments. Even in a nucleosome lacking linker DNA, i.e., in a nucleosome core particle (NCP), the H3 N-tail signals more closely resembled those of an H3 N-tail peptide (residues 1–44) bound to an H3-tailless NCP than those of the free peptide[10]. Consistent with these experimental data, MD simulations demonstrated that the H3 N-tail robustly and dynamically packs onto the core DNA[10]. In addition, interactions of both the H4 and H3 N-tails with nucleosome DNA have been reported to regulate histone-tail modifications[21]. At high salt concentrations, which weaken electrostatic interactions between the H3 N-tail and DNA, nearly all H3 N-tail signals in both the NCP and nucleosome shifted toward those of the DNA-free H3 N-tail fragment peptide[15]. These interactions are also consistent with a previous integrative study combining NMR, chemical reactivity, MD, and fluorescence analyses[19], as well as single-molecule FRET experiments[22] and MD simulations[23–28]. Interestingly, however, almost all H3 N-tail signals in the

[1]Graduate School of Medical Life Science, Yokohama City University, Yokohama, Japan. [2]Graduate School of Agriculture, Kyoto University, Sakyo-ku, Japan. [3]Laboratory of Chromatin Structure and Function, Institute for Quantitative Biosciences, The University of Tokyo, Bunkyo-ku, Japan. [4]Department of Biophysics, Graduate School of Science, Kyoto University, Sakyo-ku, Japan. [5]Department of Physical Chemistry, Faculty of Chemistry, Complutense University of Madrid, Madrid, Spain. [6]Laboratory for Epigenetics Drug Discovery, RIKEN Center for Biosystems Dynamics Research, Yokohama, Japan. [7]College of Pharmaceutical Sciences, Ritsumeikan University, Shiga, Japan. [8]Department of Computational Biology and Medical Sciences, Graduate School of Frontier Sciences, The University of Tokyo, Bunkyo-ku, Japan. [9]Department of Biological Sciences, Graduate School of Science, The University of Tokyo, Bunkyo-ku, Japan. [10]RIKEN Center for Integrative Medical Sciences, Yokohama, Japan. ✉e-mail: nisimura@yokohama-cu.ac.jp

NCP exhibited a slight but significant high-field shift relative to the corresponding signals in the nucleosome, moving even further away from the free-peptide signals in the $^1H$ and/or $^{15}N$ dimensions[10,13,14]. These observations indicate that the H3 N-tail dynamically contacts DNA with distinct interaction characteristics in the NCP and nucleosome contexts. These characteristics correlate with the markedly different rate constants of H3K14 acetylation by the histone acetyltransferase (HAT) domain of Gcn5: in the NCP, the acetylation rate of H3K14 is greatly reduced compared with that in the nucleosome[11,14].

Previously, using NMR, we examined the dynamics of the H3 N-tails in nucleosomes with and without tetra-acetylation of the H4 N-tail at K5, K8, K12, and K16 (H4-4Kac) and compared the rate of the H3 K14 acetylation. We found that H4-4Kac alters the dynamic state of the N3 N-tail, enhancing its acetylation at K14 by Gcn5. In the unmodified chromatosome, by contrast, linker histone H1.4 induces asymmetric dynamic states of the H3 N-tail. However, the asymmetric dynamic states of the H3 N-tails equilibrate on the time scale of the enzyme reaction, with the result that the acetylation rate of H3 K14 by Gcn5 in the unmodified chromatosome is comparable to that in the nucleosome. This suggests that linker histone H1 does not significantly repress acetylation of the H3 N-tail[14]. In this study, using coarse-grained molecular dynamics (CG-MD) simulations, in addition to NMR and cryo-EM methods, we have further investigated the effect

of H4-4Kac on H3 N-tail dynamics in the presence and absence of linker histone H1 to elucidate mechanisms underlying the maintenance of euchromatin and/or heterochromatin.

## Results

### Overall structure of the H4-4Kac chromatosome

We previously reported that the frequency of interaction between the histone H3 N-tail and DNA differs in the unmodified and H4-4Kac nucleosomes[11,14]. In the present study, we reconstructed the H4-4Kac chromatosome by adding the linker histone H1.4 to the H4-4Kac nucleosome. First, we performed single-particle cryo-EM experiments on the H4-4Kac chromatosome and obtained two classes of three-dimensional (3D) reconstruction density maps at resolutions of 2.92 Å and 2.89 Å (Figs. 1A and S1). Although H1.4 comprises a globular domain and two intrinsically disordered N- and C-tails, only its globular domain was successfully fitted into the density maps of the two classes (Figs. 1A and S2).

In both structures, the proximal linker-DNA to which H1.4 binds is clearly resolved, as it is in the unmodified chromatosome. In contrast, the distal linker-DNA appears collapsed with slightly different orientations between the two classes, suggesting that H4-4Kac affects the flexibility of the distal linker-DNA (Fig. 1B). It should be noted that the DNA sequences within these two H4-4Kac chromatosome structures were determined from

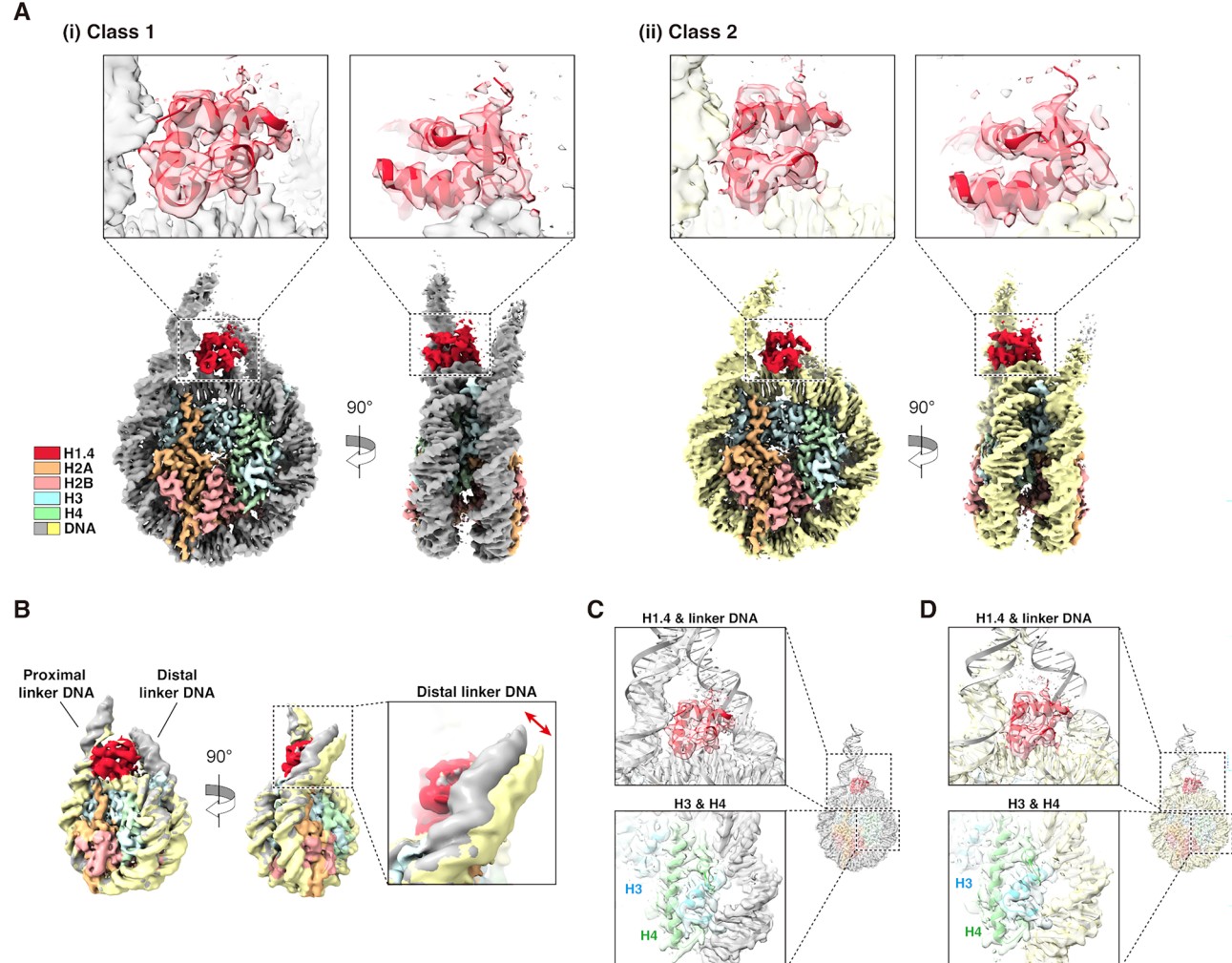

**Fig. 1 | Cryo-EM structures of H4-4Kac chromatosome. A** Class 1 (i) and class 2 (ii) cryo-EM structures of the H4-4Kac chromatosome. Close-up views of the cryo-EM density maps fitted to the structural model (PDB ID: 7K5Y) are shown above. **B** Superimposition of the two classes of cryo-EM structures of the H4-4Kac chromatosome. Both maps were low-pass-filtered to a resolution of 6.0 Å. A close-up

view of the distal linker DNA is shown on the right. **C** The class 1 cryo-EM structure with the fitted structural model (PDB ID: 7K5Y). **D** The class 2 cryo-EM structure with the fitted structural model (PDB ID: 7K5Y). In (**C**, **D**), close-up views of H1.4 (upper) and H4 (bottom) are shown on the left.

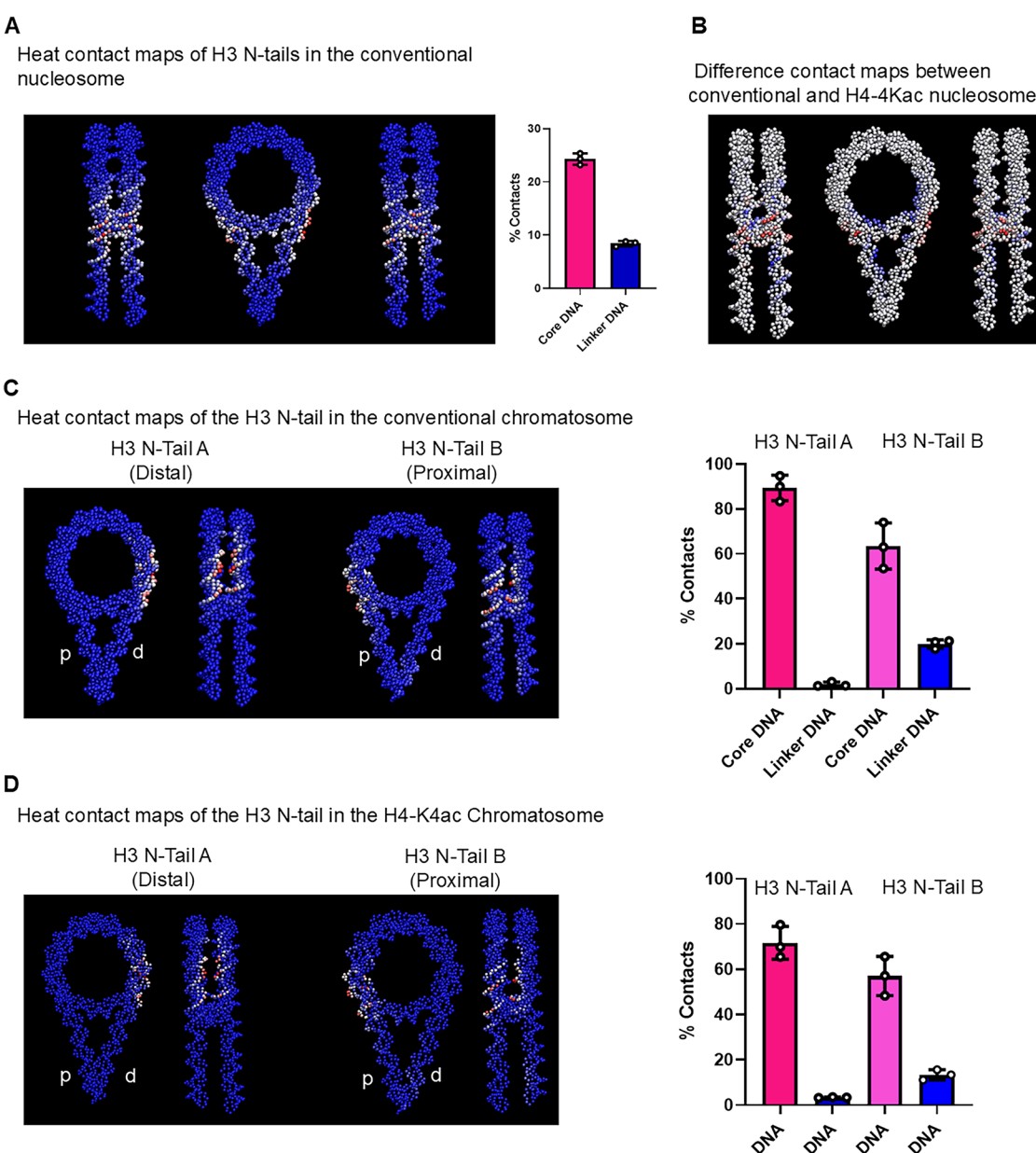

**Fig. 2 | CG-MD simulations of H3 N-tail interactions with DNA. A** Heat contact maps of the two H3 N-tails in the conventional nucleosome, showing side and top views of the nucleosome (left) and the percentage of contacts between the H3 N-tail and core- and linker-DNA (right). **B** Difference contact maps of the H3 N-tails in the conventional nucleosome and those in the H4-4Kac nucleosomes, with positive values indicated in blue and negative ones in red. Heat contact maps of the proximal H3 N-tail (N-Tail B) and the distal H3 N-tail (N-Tail A) with core-DNA, proximal linker-DNA ('p'), and distal linker-DNA ('d') in the conventional (**C**) and H4-4Kac (**D**) chromatosomes, showing top and side views of the chromatosome structures (left) and the percentages of contacts during 10 μs simulations (right). Three independent MD simulations were performed for each state ($n = 3$), and error bars are defined as the standard deviation of the three independent simulations.

the cryo-EM maps, and the DNA orientation was identical in both (Fig. S3). Although the two linker-DNAs in the H4-4Kac chromatosome are flexible and asymmetric, the core structures of the histone octamer, the globular domain of linker histone, and the core-DNA are essentially the same as those in the unmodified chromatosome (Fig. 1C, D).

**Coarse-grained molecular models of nucleosomes and chromatosomes**

To reveal the dynamics of the H3 N-tails, we performed CG-MD simulations on the conventional and H4-4Kac nucleosomes, as well as on the conventional and H4-4Kac chromatosomes with H1.4. The simulations showed that, in the conventional nucleosome, both H3 N-tails

interact with linker-DNA in addition to core-DNA (Fig. 2A and Supplementary Data 1 (cells B3 and C3)), as suggested by earlier NMR experiments. In addition, H4 N-tail extensively contacts DNA, consistent with previous NMR, cross-linking, and MD studies[16–18,21,29–31]. On H4-4Kac, the H4 N-tails are released from their contact with core-DNA (Fig. S4C and Supplementary Data 1 (lines 3–4 and 10–11)), in agreement with recent electrostatic potential analyses based on NMR data[20] and with our previous NMR results showing that acetylation increases the signal intensities of H4 N-tails, indicative of enhanced tail flexibility[14]. The contact maps of the H3 N-tail reveal subtle differences, albeit not significant, in several regions (Figs. 2B and S4B). These observations seem to correspond to our previous NMR experiments[11].

In both conventional and H4-4Kac chromatosomes containing H1.4, CG-MD indicated that the H3 N-tail exhibits asymmetric behavior: the distal H3 N-tail (N-Tail A) contacts only core-DNA, whereas the proximal H3 N-tail (N-Tail B) contacts both linker-DNA and core-DNA, as found in the conventional nucleosome (Fig. 2A and Supplementary Data 1 (cells B3 and C3)). This asymmetry arises from the binding of the linker histone H1.4 to proximal linker-DNA, in addition to the dyad axis of the nucleosome. The N- and C-tails of H1.4 fluctuate dynamically and transiently contacting either side of the linker-DNA (Fig. S5). In both conventional and H4-4Kac chromatosomes, the C-tail of H1.4 contacts both core- and distal linker-DNA with similar percentages of contacts with core- and distal linker-DNA, but makes less contacts with proximal linker-DNA, while the N-tail of H1.4 primarily contacts proximal linker-DNA without any contact to core-DNA (Fig. S5E, F and Supplementary Data 1 (in the chromatosome, cells B22 + D22 to core-DNA and E22 + C22 to linker-DNA; in the H4-4Kac one, cells B32 + D32 to core-DNA and E32 + C32 to linker-DNA)). This difference reflects the long C-tail relative to the short N-tail of H1.4. Consistently, in conventional chromatosomes, previous cryo-EM structures showed that the H1 C-tail binds primarily to distal linker DNA[5], and earlier CG-MD simulations demonstrated that the H1 C-tail dynamically associates with distal linker DNA rather than proximal linker DNA, in addition to dyad DNA[32,33]. These stochastic fluctuations of the H1.4 tails regulate the H3 N-tail contact with DNA, hindering the contacts of the distal H3 N-tail with the linker-DNA but not that of the proximal H3 N-tail in both chromatosomes. The asymmetric structures of the H3 N-tails are consistent with recent MD simulations reporting different DNA-interaction modes of the H3 N-tails in the chromatosome[28]. Furthermore, acetylation mimetics of the H3 N-tail were reported to enhance the dynamic exchange of the nucleosome-bound H1 C-tail on linker DNA, supporting the idea that the H1 C-tail modulates both the dynamics and accessibility of the H3 N-tails[34].

Although CG-MD simulations indicated that the dynamics of the H3 N-tails are very similar between the conventional and H4-4Kac chromatosomes within the 10 μs simulations applied, there are some differences: the percentages of contacts of the distal H3 N-tail (N-tail A) with core-DNA is ~90% in the conventional chromatosome and ~75% in the acetylated one; in addition, the percentages of the proximal H3 N-tail (N-tail B) contacts with core- and linker-DNA is, respectively, ~64% and ~21%, in the conventional chromatosome, and ~57% and ~13% in the acetylated one (Fig. 2C, D and Supplementary Data 1 (in the chromatosome, for H3 A cells B17 + D17 with core-DNA and C17 + E17 with linker-DNA and for H3 B B18 + D18 with core-DNA and C18 + E18 with-linker DNA; in the H4-4Kac one same but regarding lines 26-33)).

The CG-MD simulations revealed that the H4 N-tail with 4Kac is released from core-DNA into a solvent-exposed state (Fig. S5A, B and Supplementary Data 1 (lines 19 + 20 for the chromatosome and 30 + 31 for the H4-4Kac one)); this modification influences the dynamics of the H1.4 tails, thereby resulting in the slightly different H3 N-tail behavior found in both systems, as suggested by previous NMR experiments.

## H4-4Kac affects the dynamics of the histone H3 N-tails in the chromatosome

To investigate the dynamics of the H3 N-tails, we measured their [$^1$H-$^{15}$N] HSQC spectra in both the unmodified and H4-4Kac chromatosomes. The linker histone H1.4 titration experiments confirmed that the NMR samples of unmodified and H4-4Kac chromatosomes were free of unbound nucleosomes. Upon addition of the linker histone H1.4 to the H4-4Kac nucleosome, the signals of all H3 N-tail residues shifted such that they nearly overlapped with the corresponding signals observed in the nucleosome core particle (NCP), which contains 145 bp of DNA wrapped around the histone octamer without linker-DNA (Fig. 3A–C).

In our previous studies, we divided the H3 N-tail in the NCP and the nucleosome into the following subdomains based on their heteronuclear Overhauser effect values: basic segment 1 (BS1, T3–S10); linker L1 (T11–P16); basic segment 2 (R17–S28), subdivided as BS2$_1$ (R17–L20) and BS2$_2$ (A21–S28); and linker L2 (A29–K36)[11,12,14]. Here, differences in

chemical shift between the H4-4Kac chromatosome and NCP were relatively small in BS1, L1, BS2$_1$, and BS2$_2$. In the unmodified chromatosome, by contrast, residues in BS2$_2$ and L2 together with K18 and Q19 in the H3 N-tails shift toward to the corresponding signals of the NCP. In addition, doublet signals are observed for several residues in BS1 (T3, K4, Q5, T6, A7, R8, K9) and L1 (G12, K14) in the unmodified chromatosome: one closer to the corresponding NCP signal, and the other closer to the corresponding nucleosome signal. This suggests that BS1 and L1 adopt two states: an NCP-like state and a nucleosome-like state[11,14]. In the H4-4Kac chromatosome, however, no such doublets were observed; instead, the H3 N-tail signals overlapped with the corresponding NCP signals (Fig. 3C, D and Supplementary Data 2).

## Rate of histone H3 N-tail acetylation in the H4-4Kac chromatosome

Previous studies have shown that the H3 N-tails in NCP make dynamic and robust contact with core-DNA[10,12–15]. In this state, acetylation of H3 K14 by the Gcn5 HAT domain is greatly suppressed relative to that in the nucleosome, where the H3 N-tails additionally contact the linker-DNA[9,14,15], which is more accessible to the enzyme[14]. Next, therefore, we compared the rate of H3 K14 acetylation by the Gcn5 HAT domain between the unmodified and H4-4Kac chromatosomes. The signal intensity for K14 only gradually decreased while that for acetylated K14 appeared (Fig. 4A, B and Supplementary Data 3), indicating that the acetylation rate of H3 K14 in the H4-4Kac chromatosome was markedly reduced as compared with that in the unmodified chromatosome, and was nearly identical to that observed in NCP. This finding is consistent with our NMR results showing that the chemical shifts of the H3 N-tails in the H4-4Kac chromatosome closely resemble those of the H3 N-tails in NCP.

## Discussion

It has been proposed that each H4 N-tail, protruding above each core-DNA, dynamically moves on the core-DNA in a 'fuzzy' complex, fluctuating from near to dyad axis to the opposite site of NCP, independently of the linker-DNA[16,17]. On the other hand, each H3 N-tail, protruding from two DNA gyres of core-DNA, exhibits dynamic and robust fluctuations on rigid core DNA in NCP[10,12–15], and additionally interacts with flexible linker-DNA in the nucleosome[9,11,14,15]. Thus, the rate of H3 K14 acetylation is much slower in NCP than in the nucleosome[14] (Fig. 4B).

Here, in the case of the H4-4Kac nucleosome, two distinct H3 N-tail NMR signals were observed for the BS2$_2$ region, along with an increased rate of H3 K14 acetylation relative to the unmodified nucleosome. We previously proposed that two different DNA contact states exist in the H4-4Kac nucleosome[11,14]. Our CG-MD simulations showed small but subtle difference in the H3 N-tail contacts with core- and linker-DNA between the conventional and H4-4Kac nucleosomes, which might be related to the different NMR signals (Figs. 3B and S4B)[11].

In the unmodified chromatosome, the core domain of the linker histone H1.4 is located at the dyad axis of core-DNA[6]. A previous study of the H3 N-tail as a free peptide, and in the nucleosome and chromatosome showed that the H3 N-tail transiently and electrostatically contacts nucleosome DNA, and its dynamics are reduced by the binding of linker histone H1, depending on the C-terminal domain of H1[9]. Consistent with previous CG-MD simulations[32,33], our CG-MD simulations showed that the H1 C-tail contacts core-DNA in addition to the distal linker-DNA, while the H1 N-tail contacts the proximal linker-DNA and has less contact with core-DNA (Fig. S5). This leads to an asymmetric structure in which one of the H3 N-tails is prevented from binding the distal linker-DNA and instead dynamically contacts core-DNA, while the other H3 N-tail dynamically contacts the linker-DNA located proximal to H1. Thus, CG-MD simulations showed that the distal H3 N-tail contacts primarily core-DNA, while the proximal H3 N-tail contacts both linker-DNA and core-DNA. Accordingly, the H3 N-tail in the unmodified chromatosome exhibits both NCP-like and nucleosome-like NMR signals, corresponding to two distinct contact states of the H3 N-tail; however, the equilibrium between the two

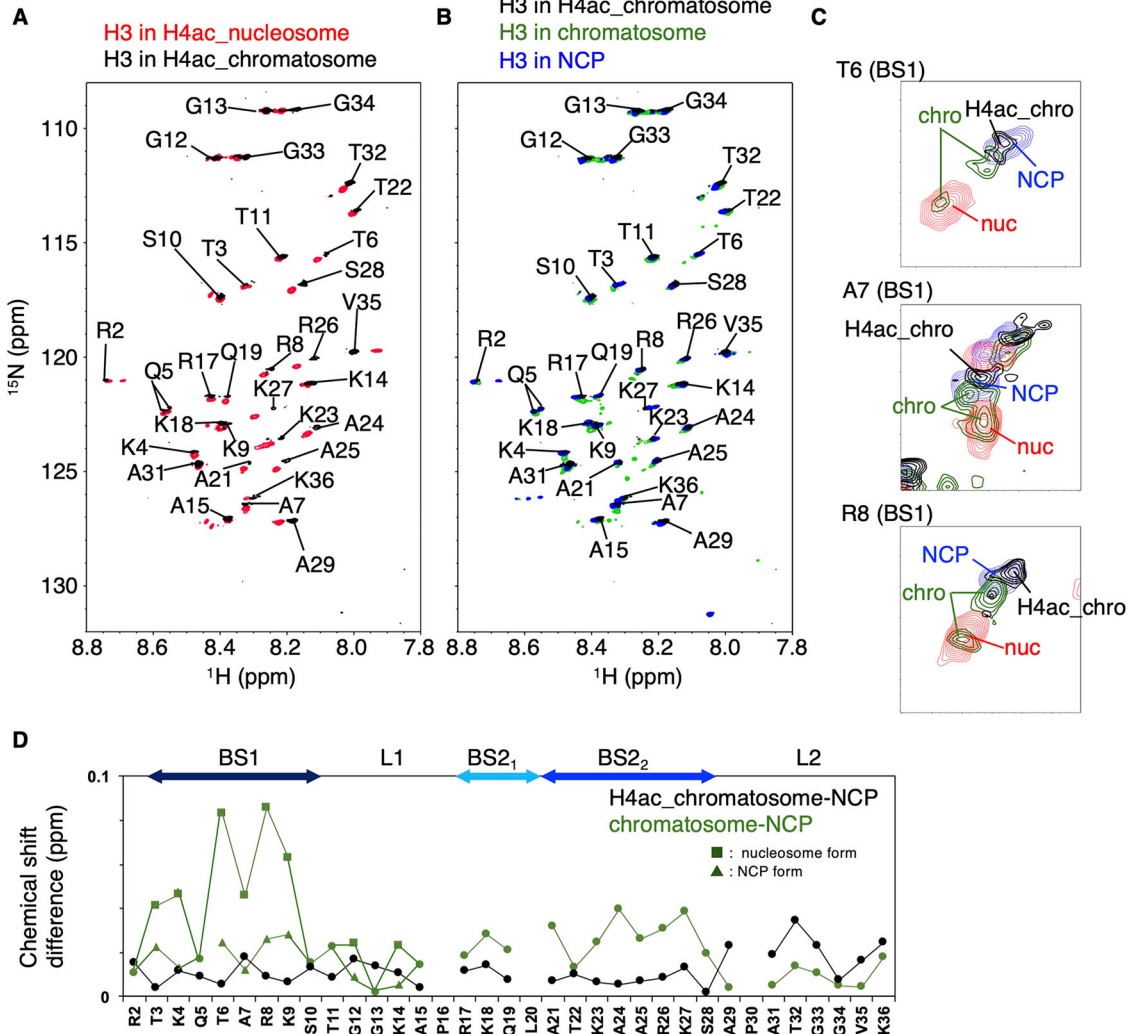

**Fig. 3 | NMR studies of the H3 N-tails in the H4-4Kac nucleosome.**
**A** Superposition of $^1$H–$^{15}$N HSQC spectra of the H3 N-tails in the H4-4Kac nucleosome (red) and the H4-4Kac chromatosome (black) at 25 mM NaCl. **B** Superposition of $^1$H–$^{15}$N HSQC spectra of the H3 N-tails in the H4-4Kac chromatosome (black), the unmodified chromatosome (green), and NCP (blue) at 25 mM NaCl. **C** Expanded spectra of three residues of the H3 N-tails in the H4-4Kac chromatosome (black), unmodified chromatosome (green), unmodified nucleosome (red), and NCP (blue) at 25 mM NaCl. **D** Chemical shift differences of the H3 N-tail residues between NCP and the H4-4Kac chromatosome (black), and between NCP and the unmodified chromatosome (green) at 25 mM NaCl. Symbols indicate a singlet signal (filled circle), NCP side of a doublet signal (filled triangle), and nucleosome side of a doublet signal (filled square).

states caused by asymmetric binding of H1 seems to be faster than the enzyme reaction. Thus, the rate of H3 K14 acetylation in the unmodified chromatosome is similar to that in the unmodified nucleosome[14]. This is consistent with previous MD simulations demonstrating that the globular domain of H1 in the chromatosome has a highly dynamic character[32,35,36], and is also supported by fluorescence recovery after photobleaching experiments showing that H1 remains associated with chromatin for several minutes[37]. The location of H1 fluctuates between the two linker-DNAs on the timescale of the enzyme reaction, thereby allowing H3 N-tails to alternately contact the linker-DNA, where it is accessible to enzyme.

In the H4-4Kac chromatosome, only NMR signals corresponding to the NCP-like H3 N-tails were observed; this suggests that H4-4Kac accelerates the dynamic exchange of H1 between the two linker-DNAs, preventing the H3 N-tails from contacting linker-DNA on the NMR timescale (Fig. 4C). The Cryo-EM structures of the H4-4Kac chromatosome revealed that the core configuration of the globular domain of H1.4, core-histones, and nucleosomal DNA is essentially identical to that of the unmodified chromatosome, regardless of the fluctuation timescale of the linker histone H1.4, which is on the order of either a few seconds for NMR measurements in the H4-4Kac chromatosome, or several tens of minutes for enzyme

reactions in the unmodified chromatosome. In the current CG-MD simulations, both conventional and H4-4Kac chromatosomes showed that the distal H3 N-tail contacts core-DNA, while the proximal H3 N-tail contacts linker-DNA in addition to core-DNA. The simulations indicated that the distal H3 N-tail interacts with core-DNA less in the H4-4Kac chromatosome (75%) than in the conventional one (90%), and the proximal H3 N-tail contacts with linker-DNA less in the H4-4Kac chromatosome (13%) than in the conventional one (21%) (Figs. 2C, D and 4C). This may relate to the different binding dynamics of H1 locations in the conventional and H4-4Kac chromatosomes. In addition, it has been previously reported that acetylation mimetics of the H3 N-tail enhance dynamic exchange of the nucleosome-bound H1 C-tail on linker-DNA, supporting the notion that the H1 C-tail can modulate both the dynamics and accessibility of H3 N-tails[34].

The acetylated H4 N-tail is released from core-DNA, which is dynamically occupied by the unmodified H4 N-tail in the fuzzy complex (Fig. S5A, B). As a result, the BS2$_2$ region of the H3 N-tail is brought into dynamic contact with the region of core-DNA previously occupied by the H4 N-tail. Thus, the change in H3 N-tail binding state due to H4-4Kac may be either causing the linker histone H1 to dynamically and rapidly fluctuate

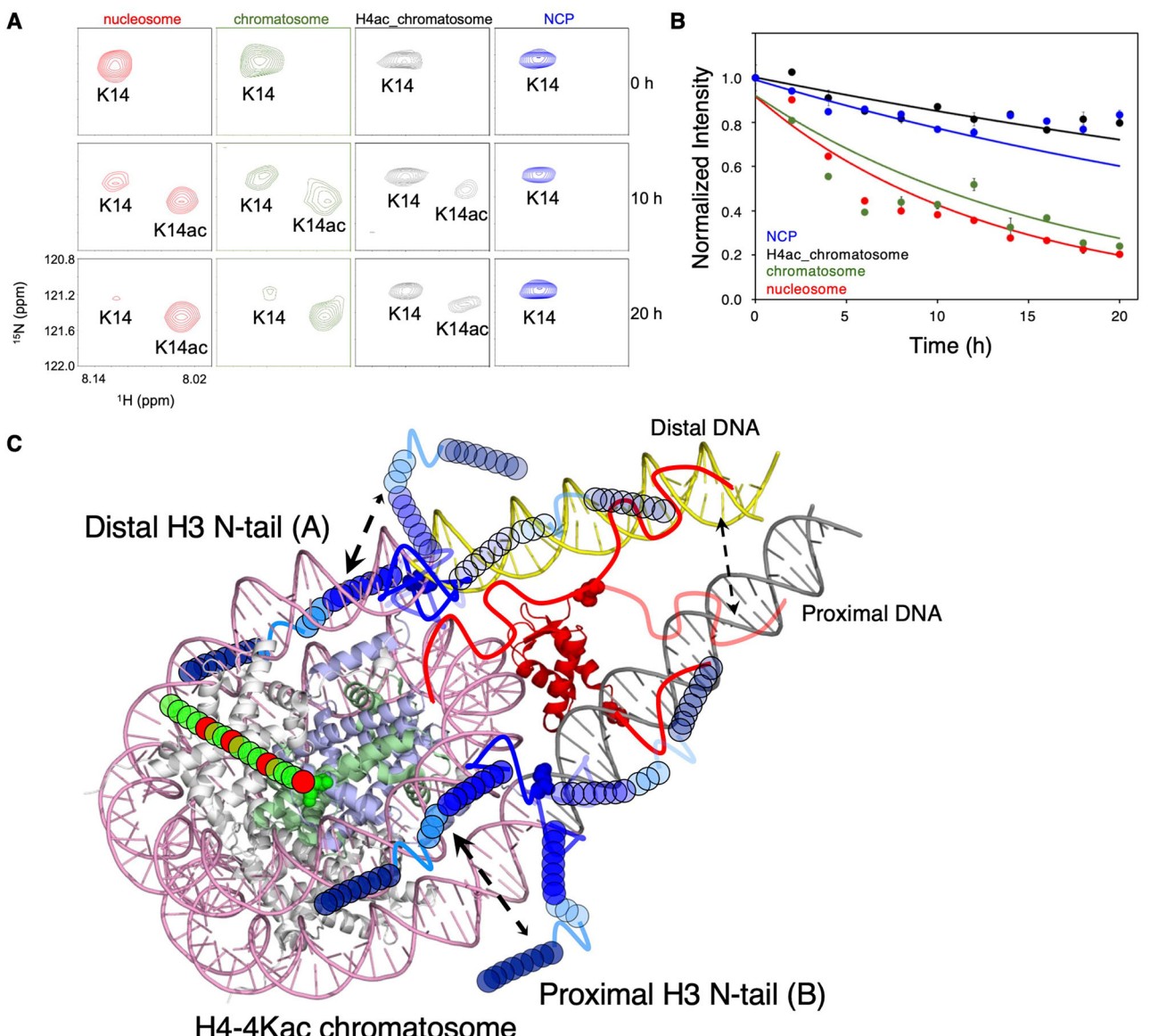

**Fig. 4 | Effect of H4-4Kac on the H3 N-tail acetylation rate and model of H3 N-tail–DNA interactions in the H4-4Kac chromatosome. A** Time-dependent signal changes of K14 and acetylated K14 of the H3 N-tail in the nucleosome (red), chromatosome (green), H4-4Kac chromatosome (black), and NCP (blue) after the addition of Gcn5. **B** Comparison of the rate of Lys14 acetylation in the nucleosome (red), chromatosome (green), H4-K4ac chromatosome (black), and NCP (blue) by time-resolved NMR spectroscopy. Signal intensities were normalized by the initial signal measured before the addition of Gcn5. Lines represent fits to a single-exponential equation ($n = 1$ independent experiment). Error bars indicate the uncertainty of the exponential fit estimated by Monte Carlo analysis in GLOVE (100 iterations) and do not represent variability from experimental replicates. The apparent rate constants are summarized in Table 2. **C** Model of DNA interaction with the H3 N-tail (blue) in H4-4Kac chromatosome (Protein Data Bank ID code 7K5Y). In the H4-4Kac chromatosome, the H3 N-tails adopt an NCP-like form due to the binding of H1.4. In this case, two binding states—on the distal linker-DNA side (yellow) and on the proximal linker DNA side (gray)—exist in dynamic equilibrium.

on DNA, or accelerating the binding and dissociation rates of H1.4 on DNA, both processes occurring on the timescale of NMR measurements. This would lead to rapid changes in the asymmetric positioning of H1 between two linker-DNAs. Consequently, the interaction of the H3 N-tail with linker-DNA is prevented in the H4-4Kac chromatosome, and both H3 N-tails dynamically contact core-DNA, resulting in only NCP-like signals (Figs. 3 and 4C).

In summary, the present study has demonstrated that acetylation of the H4 N-tail modulates linker histone H1 binding, thereby enhancing the interaction between the H3 N-tail and core-DNA within the nucleosome and consequently inhibiting acetylation of the H3 N-tail. This mechanism may contribute to the structural transition from euchromatin to hetero-chromatin. Linker histone H1 has been reported to bind to Histone

Deacetylase Complex 1 protein in *Arabidopsis*[38]. It will be important in future studies to clarify how linker histone H1 binding contributes to deacetylation.

## Methods

### Preparation of histones H2A, H2B and H3

Recombinant human histone H2A and H2B proteins were produced in *Escherichia coli* BL21 cells. $^2$H/$^{15}$N-labeled histone H3.1 proteins were expressed in 100% deuterated M9 minimal medium containing $^{15}$N-ammonium chloride[11]. H2A, H2B, and $^2$H/$^{15}$N-labeled H3.1 proteins were insoluble. The precipitates were solubilized in 20 mM Tris-HCl buffer (pH 7.5) containing 7 M guanidine-HCl and 10 mM DTT, loaded onto a Superdex 200 26/60 gel filtration column (Cytiva), and separated using

20 mM sodium acetate buffer (pH 5.2) containing 7 M urea, 1 M NaCl, 1 mM EDTA, and 5 mM 2-mercaptoethanol as the elution buffer. The histone proteins were subjected to ion-exchange chromatography on a HiTrap SP 5 ml HP column (Cytiva) and then dialyzed against distilled water. The histone protein solutions were lyophilized and used for nucleosome[11].

## Preparation of histone H4

Recombinant human histone H4 protein containing K5/K8/K12/K16 acetylation (H4-4Kac) was synthesized by genetic code reprogramming in a coupled transcription–translation cell-free system[39]. pCR2.1 with cDNA containing human histone H4 ORF with or without the codons of the K5/K8/K12/K16 residues replaced with the TAG triplets and a terminal TAA stop codon[40], were used as the template in a coupled transcription–translation cell-free system with a 9-ml reaction solution dialyzed against a 90-ml external feeding solution. The 9-ml reaction solution contained 0.37 volume of the low-molecular-weight creatine phosphate tyrosine (LMCPY) mixture [160 mM HEPES-KOH buffer (pH 7.5), containing 4.1 mM l-tyrosine, 3.5 mM ATP, 2.4 mM each of GTP, CTP, UTP, 0.22 mM folic acid, 1.8 mM cAMP, 74 mM ammonium acetate, 210 mM creatine phosphate, 5 mM DTT, 530 mM potassium l-glutamate, and 11% PEG8000], 0.075 volume of the 19-amino acid mixture [20 mM each of the amino acids other than l-tyrosine], 0.01 volume of 17.5 mg/ml tRNA, 0.26 volume of *E. coli* S30 extract from RFzero strains[39–41], 14 mM Mg(OAc)$_2$, 10 mM acetyllysine, 5 mM nicotinamide, 0.05% NaN$_3$, 250 µg/ml creatine kinase, 8 µM pyrrolysine-specific tRNA (tRNA$^{Pyl}$)[39,40], 8 µM pyrrolysyl-tRNA synthetase (PylRS) mutant KacRS_6mt[39,40], 67 µg/ml T7 RNA polymerase, and 16–64 µg/ml template plasmid. The 90-ml external feeding solution contained 0.03 volume of 10×S30 buffer [100 mM Tris-acetate buffer (pH 8.2), containing 600 mM potassium acetate, 160 mM Mg(OAc)$_2$, and 10 mM DTT], 0.37 volume of the LMCPY mixture, 0.075 volume of the 19-amino acid mixture, 14 mM Mg(OAc)$_2$, 20 mM acetyllysine, 10 mM nicotinamide, and 0.05% NaN$_3$[39]. Cell-free synthesis in the dialysis mode was performed at 37 °C for 6–18 h, using 30 cm$^2$ of dialysis membrane per ml reaction solution[39]. The reaction mixture was centrifuged for 30 min at 30,000 × *g* at 4 °C, and the protein precipitates were solubilized in 50 mM Tris-HCl buffer (pH 8.0) containing 6–7 M guanidine-hydrochloride and 10 mM DTT[39]. The samples were centrifuged for 30 min at 30,000 × *g* at 4 °C, and the supernatants were filtered through 0.45 µm and 0.22 µm pore size filters (Merck Millipore). The samples were loaded onto a HisTrap HP column (Cytiva) and eluted with an imidazole gradient from 20 mM to 500 mM under 50 mM Tris-HCl buffer (pH 8.0) containing 500 mM NaCl, 5% glycerol, and 7 M urea, and the eluates were dialyzed three times against cold distilled water containing 5 mM 2-mercaptoethanol for 2 h at 4 °C[39]. The N-terminal N11 histidine-rich affinity tag followed by a TEV protease recognition site (EHLYFQ) was cleaved by incubation with TEV protease at a molar ratio of histone:TEV protease = 5–10:1 at 4 °C overnight[39]. The solution of cleaved histone H4 protein containing 7 M urea was then reapplied to a HisTrap HP column (Cytiva), and the flow-through fraction was collected[39]. The histone H4 protein samples were subjected to ion-exchange chromatography on a Mono S 10/100 GL column (Cytiva, 17516901), dialyzed four times against cold distilled water for 3 h at 4 °C, and lyophilized[39].

## Nucleosome reconstitution

Histone H2A, H2B, H3, and H4-4Kac proteins were mixed at an equi-molar ratio in 10 mM Tris-HCl buffer (pH 7.6) containing 7 M guanidine-hydrochloride and 5 mM DTT and dialyzed four times against 10 mM Tris-HCl buffer (pH 7.6) containing 2 M NaCl and 5 mM DTT for 3 h at 4 °C[39]. Histone octamers were purified by gel filtration chromatography using a HiLoad Superdex 200 16/60 column (Cytiva, 28-9893-35) under 10 mM Tris-HCl buffer (pH 7.6) containing 2 M NaCl and 5 mM DTT and concentrated in Amicon Ultra-15 centrifugal filter units (Merck Millipore, 30 kDa MWCO). Histone octamers and the 193-bp DNA were mixed at a 1:1.1 molar ratio in 10 mM Tris-HCl buffer (pH 7.6) containing 2 M KCl, 1 mM EDTA, and 5 mM DTT, and the solution was dialyzed against the same buffer for 4 h at 4 °C. The concentration of KCl was gradually decreased by diluting for 30 h with the buffer containing 250 mM KCl using a peristaltic pump (ATTO, SJ-1211II-H)[39]. The sample was centrifuged for 10 min at 15,000 × *g* at 4 °C, and the supernatant containing the reconstituted nucleosomes was incubated at 55 °C for 2 h and stored at 4 °C[39]. Reconstituted nucleosomes were purified on a 6% native polyacrylamide gel (acrylamide:N,N′-methylenbisacrylamide = 59:1), using a Model 491 Prep Cell apparatus (Bio-Rad, 1702928), and the eluted fractions were concentrated in Amicon Ultra-15 centrifugal filter units (Merck Millipore, 10 kDa MWCO)[39].

## Purification of human histone H1.4

The His$_6$-SUMO-tagged H1.4 protein was produced in the *Escherichia coli* BL21 (DE3) strain, which carried the minor tRNA expression vector (Codon(+)RIL; Agilent). The cells producing His$_6$-SUMO-tagged H1.4 were resuspended in buffer (20 mM Tris-HCl (pH 7.6), 1 mM DTT, 1 M NaCl, and 10% glycerol) and disrupted by sonication. The cell lysate was clarified by centrifugation, and the supernatant was applied to a HisTrap FF column (Cytiva), The column was washed with buffer containing 20 mM Tris–HCl (pH 7.6), 1 M NaCl, 1 m DTT, and 2 mM imidazole, and eluted with a gradient of imidazole (2–400 mM). Fractions containing His$_6$-SUMO-tagged H1.4 were collected, and the His$_6$-SUMO segment was removed by HRV3C Protease (Fuji Film) during dialysis against buffer containing 20 mM Tris-HCl (pH 7.5), 100 mM NaCl, 1 mM DTT, 1 mM EDTA, and 10% glycerol. After tag removal, the sample was loaded onto a HiTrap SP column (Cytiva) equilibrated with 20 mM Tris–HCl (pH 7.6), 50 mM NaCl, 1 mM DTT, and eluted with a NaCl gradient (50–1000 mM). Protein fractions were dialyzed against buffer containing 25 mM MES (pH 6.0), 25 mM NaCl, 2 mM DTT, and concentrated using an ultrafiltration cartridge (Millipore).

## 193 bp DNA sequence for nucleosome

5′-ATCGGACCCTATCGCGAGCCAGGCCTGAGAATCCGGTGCCG
AGGCCGCTCAATTGGTCGTAGACAGCTCTAGCACCGCTTAAA
CGCACGTACGCGCTGTCCCCCGCGTTTTAACCGCCAAGGGGAT
TACTCCCTAGTCTCCAGGCACGTGTCAGATATATACATCCAGG
CCTTGTGTCGCGAAATTCATAGAT-3′

 5′-ATCTATGAATTTCGCGACACAAGGCCTGGATGTATATA
TCTGACACGTGCCTGGAGACTAGGGAGTAATCCCCTTGGCGG
TTAAAACGCGGGGGACAGCGCGTACGTGCGTTTAAGCGGTG
CTAGAGCTGTCTACGACCAATTGAGCGGCCTCGGCACCGG
ATTCTCAGGCCTGGCTCGCGATAGGGTCCGAT-3′

## NMR experiments

NMR experiments were performed on AVANCE 600-MHz and AVANCE III HD 950-MHz spectrometers with a triple-resonance TCI cryogenic probe (Bruker Bio Spin) at 298 K using 20-80 µM samples dissolved in 25 mM MES (pH 6.0), 1 mM DTT, and 5% D$_2$O. The unlabeled linker histone H1.4 was added step-by-step to reach ratios of 0.15, 0.3, 0.5, 1.0, and 1.5 to the 40 µM nucleosome containing $^{15}$N/$^2$H-labeld H3. In each step, any precipitate that formed during the titration process was removed by centrifugation, and the $^1$H−$^{15}$N HSQC spectrum of the supernatant of the nucleosome/H1.4 solution was acquired at 298 K. In the $^1$H−$^{15}$N HSQC spectrum with 1.5 equivalents of H1.4 (60 µM) added, we confirmed that there were no nucleosome signals, indicating all the nucleosomes in the NMR solution were in the H1-bound form (i.e., chromatosome). In addition, we performed electrophoretic mobility shift assays of H1.4 binding to each of the conventional and H4 K4ac nucleosomes to monitor the corresponding chromatosome formation (Fig. S6). NMR data were processed by NMRPipe[42], and analyzed by NMRViewJ ([43]: One Moon Scientific, Inc., Westfield, NJ, USA), and PINT[44]. Averaged chemical shift differences were calculated by the equation [(ΔHN)$^2$ + (ΔN/5)$^2$]$^{1/2}$, where ΔHN and

**Table 1 | Cryo-EM data collection and refinement statistics**

| | #1 H4-4Kac chromatosome Class 1 (EMDB-64638) | #2 H4-4Kac chromatosome Class 2 (EMDB-64639) |
|---|---|---|
| Data collection and processing | | |
| Magnification | x81,000 | x81,000 |
| Voltage (kV) | 300 | 300 |
| Electron exposure (e–/Å²) | 59.6 | 59.6 |
| Defocus range (μm) | −1.0 to −2.5 | −1.0 to −2.5 |
| Pixel size (Å) | 1.06 | 1.06 |
| Symmetry imposed | C1 | C1 |
| Initial particle images (no.) | 11,727,117 | 11,727,117 |
| Final particle images (no.) | 358,572 | 274,548 |
| Map resolution (Å) | 2.92 | 2.89 |
| FSC threshold | 0.143 | 0.143 |
| Map resolution range (Å) | 2.76–7.42 | 2.71–6.96 |
| Refinement | | |
| Map sharpening B factor (Å²) | −23.8643 | −21.0211 |

**Table 2 | Rate constant for acetylation of K14**

| | s$^{-1}$ |
|---|---|
| nucleosome | $4.3 \times 10^{-5} \pm 7.0 \times 10^{-8}$ |
| chromatosome | $1.7 \times 10^{-5} \pm 4.1 \times 10^{-8}$ |
| H4-4Kac chromatosome | $4.5 \times 10^{-6} \pm 5.8 \times 10^{-8}$ |
| NCP | $6.9 \times 10^{-6} \pm 3.8 \times 10^{-7}$ |

$\Delta N$ are chemical shift differences of the amide proton and nitrogen atoms, respectively.

### NMR real-time monitoring of acetylation reaction
Acetylation of the chromatosome (20 μM) with acetylated H4 was carried out in 25 mM NaPB (pH6.8), 25 mM NaCl, 2 mM DTT, and 5% D2O with 250 μM acetyl-CoA and 100 nM Gcn5 (Enzo Life Sciences) at 303 K. Modification reactions were monitored by $^{1}$H-$^{15}$N TROSY-HSQC NMR experiments. The signal intensities obtained were fitted to the exponential equation by using the program GLOVE[45].

### Preparation of cryo-EM sample
H4-4Kac nucleosome (1.13 μM) was mixed with H1.4 (1.69 μM) in MES buffer [25 mM MES (pH 6.0), 25 mM NaCl, 2 mM DTT] and incubated at 310 K for 30 min. The sample was purified by GraFix method[46]. The sucrose and crosslinker gradient buffer was prepared using sucrose buffer [20 mM HEPES-KOH (pH 7.5), 50 mM KOAc, 0.2 μM Zn(OAc)$_2$, 0.1 mM TCEP-HCl (pH 7.0), 10% sucrose] and modified sucrose buffer containing 25% sucrose and 0.1% glutaraldehyde by using a Gradient Master (SK BIO International). After centrifugation, fractions containing the H4-4Kac nucleosome-H1.4 complex were collected. Finally, the sample buffer was exchanged to cryo-EM buffer [20 mM HEPES-NaOH (pH 7.5), 0.2 μM Zn(OAc)$_2$, 0.1 mM TCEP-HCl], and the purified sample was concentrated. The cryo-EM sample (2.5 μL) of the H4-4Kac nucleosome–H1.4 complex was applied to the Quantifoil holey carbon grid (Cu, R 1.2/1.3, 200-mesh) at 289 K and 100% humidity, and plunged into liquid ethane using a Vitrobot Mark IV (Thermo Fisher Scientific).

### Cryo-EM data collection and image processing
The cryo-EM dataset of the H4-4Kac chromatosome was collected at 300 kV using a Krios G4 (Thermo Fisher Scientific) and EPU software. All movies were captured at pixel size of 1.06 Å by using a K3 BioQuantum direct electron detector (Gatan). Image processing of cryo-EM data was performed using RELION 3.0[47,48] and 4.0[49]. Total 13,255 movies were corrected for drift by MOTIONCOR2[50]. The CTF estimation was performed by CTFFIND4[51]. From 12,372 corrected micrographs, 11,727,117 particles were picked and subjected to two rounds of 2D classifications. Using initial 3D model function of RELION, the initial model for the first 3D classification was created. From the results of 3D classification, two classes containing H4-4Kac nucleosome-H1.4 complex were selected and subjected to focused classifications using H1 masks. After two rounds of focused classifications, two classes containing the resolved H1.4 density maps were refined and post-processed. The resolution of each final map was 2.92 Å (class 1) and 2.89 Å (class 2), as estimated by the gold standard Fourier shell correlation (FSC) = 0.143[48]. Details of cryo-EM data collection and image processing of the H4-4Kac chromatosome are shown in Table 1.

### Visualization of cryo-EM structure
All representations of cryo-EM density maps and structural models were created by using Chimera[52] and ChimeraX[53].

### Coarse-grained molecular models and simulations
We employed a coarse-grained (CG) modelling approach to construct chromatosome structures. In this framework, each DNA nucleotide was represented by three beads corresponding to the base, sugar, and phosphate groups. The 3SPN.2C potential[54] was used to maintain B-form DNA geometry, reproduce sequence-dependent curvature, and accurately capture the persistence length and melting temperature of double-stranded DNA. For histones, each amino acid was represented as a single bead located at the $C_\alpha$ atom. Inter-residue interactions in the structured core regions were modeled using the AICG2+ potential[55], which stabilizes a given reference conformation while allowing native fluctuations. In contrast, interactions within intrinsically disordered regions, such as histone tails, were described by a flexible local potential[56] and the HPS model[57], which has been widely applied in studies of disordered proteins[58,59]. This combination of protein and DNA models, which has been extensively validated[60–64], reproduces both structural fidelity and protein–DNA binding specificity. Excluded volume and electrostatic interactions were included for intermolecular forces. Electrostatics were treated using Debye–Hückel theory. Partial charges on protein surface beads were assigned using the RESPAC algorithm[65], which reproduces the electrostatic potential of all-atom structures. For intrinsically disordered regions, where RESPAC cannot be applied, unit charges were assigned to charged residues (+1 e for lysine and arginine, −1 e for aspartic and glutamic acids). Phosphate groups on DNA carried a charge of −0.6 e for intra-DNA interactions to account for counterion condensation along the backbone, but were set to −1.0 e for protein–DNA interactions to represent counterion release upon complex

formation. Hydrogen-bonding potentials[61] were included between amino acid and phosphate pairs forming histone–DNA contacts. The reference structure of the chromatosome was prepared using cryo-EM coordinates (PDB ID: 7K5Y). Unresolved intrinsically disordered regions (i.e., histone tails) were reconstructed using Modeller[66]. A 197-bp double-stranded DNA molecule was generated with 3DNA[67]. Each molecular dynamics simulation contained one complete chromatosome. Langevin dynamics was used to integrate the equations of motion with a time step of 0.2 CafeMol time units ($\approx$0.01 ps). Simulations were performed in the *NVT* ensemble at 300 K with a monovalent salt concentration of 150 mM using OpenCafeMol[68]. For each condition, three independent simulations were conducted for $1 \times 10^9$ steps ($\approx$10 $\mu$s). Trajectory analysis was performed using the MDAnalysis library[69,70] to calculate histone tail–DNA contact frequencies and other structural parameters. Simulation structures were visualized with Visual Molecular Dynamics (VMD)[71].

### Statistics and reproducibility
Three independent MD simulations were performed for each state ($n = 3$). Cryo-EM and NMR experiments were conducted once ($n = 1$ independent experiment), and no replicate-based statistical analysis was performed.

### Reporting summary
Further information on research design is available in the Nature Portfolio Reporting Summary linked to this article.

### Data availability
Cryo-EM structures are deposited in wwPDB under the accession numbers EMD-64638 (H4-4Kac chromatosome Class 1) and EMD-64639 (H4-4KAc chromatosome Class 2). The chemical shifts are deposited in the Biological Magnetic Resonance Data Bank (BMRB) under the accession number 53182 (H4ac_chromatosome). Uncropped and unedited image of gel of Supplementary Fig. S6 is provided in Supplementary Fig. S7. All data relating to this study are presented in the manuscript and supplementary materials.

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

## Acknowledgements

This work was supported, in part, by NMR Platform (grant no. JPMXS0450100021 to Y.N.) from the Ministry of Education, Culture, Sports, Science and Technology (MEXT), Japan; by a Platform Project for Supporting Drug Discovery and Life Science Research (Basis for Supporting

Innovative Drug Discovery and Life Science Research; BINDS) from the Japan Agency for Medical Research and Development (AMED; grant nos. JP24ama121009 to H.K., JP24ama121002 to Y.T., JP21am0101073 and JP22ama121001 to Y.N.); and by the Japan Society for the Promotion of Science (JSPS) Grants-in-Aid for Scientific Research (JP21H05756 and JP23H04289 to A.F., JP22KJ0871 to K.E., JP22K06098 to Y.T., JP23H05475, JP24H02319, and JP24H02328 to H.K., JP23H02426 and JP23K27119 to Y.N.); Japan Science and Technology Agency (JST; ERATO; JPMJER1901, CREST; JPMJCR24T3 to H.K.).

## Author contributions
Conceptualization, A.F., T.U., H.K., and Y.N.; Sample Preparation, A.F., K.E., Y.Tsunaka, M.W., H.O., and Y.Takizawa; Investigation, A.F. and Y.Takizawa; MD simulation, S.B. and T.T.; Writing—Original Draft, A.F., Y.Takizawa, T.U., and S.B.; Writing—Review and Editing, H.K. and Y.N.; Funding Acquisition, A.F., H.K., and Y.N.; Resources, A.F., H.K., and Y.N.; Supervision, Y.N.

## Competing interests
The authors declare no competing interests.
