## [Transparent Peer Review file · Communications Biology]

Linker histone H1 represses H3 tail acetylation induced by H4 tail acetylation and alters its dynamics

Corresponding Author: Dr Yoshifumi Nishimura

Version 0:

Reviewer comments:

Reviewer #1

(Remarks to the Author)

This work by Furukawa et al, using cryo-EM and NMR to study how H4 N-tail (H4-4Kac) affects H3 N-tail acetylation in the context of H1.4 chromatosome. In unmodified chromatosomes, H3 N-tails dynamically interact with both core-DNA and linker-DNA, allowing acetylation. In H4-4Kac chromatosomes, H4 tail acetylation changes the dynamics, leading to H3 N-tails predominantly interacting with core-DNA between two DNA gyres and repressing its acetylation. This work suggests that acetylation of H4 N-tail modulates H1 binding, enhancing interaction between H3 N-tail and core-DNA, and thus inhibiting H3 acetylation by the Gcn5 HAT domain. Overall, the results are clearly presented and discussed. I have a few suggestions below.

1. Can the DNA base pair sequence be distinguished in the high resolution class 1 and 2 cryo-EM maps? Are class 1 and class 2 the same or not in terms of DNA sequence orientation? In the unmodified H1.4 chromatosome structure (7k5y), the DNA sequence can be assigned based on the cryo-EM density map. This point is important and could enhance the model that the author presented in Figure 3d.
2. In Figure 3a, for the NCP column, the K14ac peak doesn't show up at 10h or 20h, what causes the intensity decrease of H3K14 in NCP? I would also suggest quantifying time-dependent signal changes of H3K14ac, and list the acetylation rate constant for the exponential fittings. Are these rates relevant to in vivo histone acetylation?
3. The study indicates that it is the core domain of the linker histone H1.4 that causes the decrease in H3K14 acetylation induced by H4-4Kac, however, the exact mechanisms by which H1.4 influences the dynamics and accessibility of H3 N-tails are not clear. Understanding the specific contributions of the N- and C-terminal domains of H1.4 could further clarify the conclusion. For example, express H1.4 with N-terminal or C-terminal deletion, or both, then compare how these changes affect H3K14 tail acetylation.

Minor comments:

Figure 3a, H4ac_chromatosome column, should the NMR peak be shown in black color for consistency?

Reviewer #2

(Remarks to the Author)

In this manuscript cryoEM and NMR spectroscopy are utilized to interrogate the effects of the H1.4-chromatosome structure and H4 tail acetylation on the acetylation of the H3 tail by Gcn5. The study is timely as there is a growing interest in how the nucleosome regulates histone tail accessibility and how variations within further alter this. Overall, the results are sound and informative with respect to cross talk between H1.4 and the H3 tail as well as between the H3 and H4 tails. However, the authors overinterpret their NMR data with respect to molecular models of the tails themselves. In particular, there are a number of statements regarding the residence of the H3 and H4 tails on the double-helical gyres of the nucleosome core, the dyad, or the linker DNA that are not supported by data. The chemical shifts observed cannot be assigned to a particular structural model without more data. Though the authors can assign states (e.g. NCP-state, nucleosome-state, etc.) they cannot interpret at the molecular level where the tail is residing without additional data. A revised manuscript in which these statements are walked back would be of great interest to the community. Below are examples of statements that need to be rephrased. This list is representative but not inclusive and the full text should be revisited in this light. In addition, I have one

question regarding Gcn5 activity.

- 1) The statement “Specifically, the dynamic linker-DNA contact state (hereafter “L-state”) is switched to a more specific dynamic dyad axis DNA and linker-DNA contact state (hereafter “Ld-state”), which is related to the formation of euchromatin (11).” is not supported by the publication referenced. While the authors previously showed changes to the conformational ensemble of the H3 tail upon acetylation of the H4 tail there was no data to support specific residence time on either linker or dyad. The authors should not rephrase their previous findings in this way without a thorough reanalysis and/or additional data to support their statements.
- 2) It is unclear how a “shift toward to the corresponding signals of the NCP” indicates “increased DNA contact”. The authors provide no explanation or evidence for this.
- 3) The authors note “Previous studies have shown that the H3 N-tails in NCP make dynamic and robust contact with two gyres of core-DNA”. However, these studies also show that the H3 tails can contact the dyad in the nucleosomal context. Thus interpretation of the H3 N-tails being entirely two gyre bound in the NCP does not appear warranted.
- 4) The authors propose in the discussion that the H3 tail in the H4-4Kac nucleosome will contact the dyad as the H4 tail acetylation has displaced it from the dyad. However, I am unclear where the concept that the H4 tail is at the dyad comes from. Please cite the literature or data that indicates that the unmodified H4 tail can contact the dyad.
- 5) The authors should comment on how the presence of linker DNA or the H4 tail may inherently effect Gcn5 activity outside of H3 tail accessibility.

Reviewer #3

(Remarks to the Author)

The manuscript builds on two previous studies by the same research group, where NMR was used to characterize the conformations of the histone H3 N-terminal tail (H3 N-tail) and its interplay with tetra-acetylation of histone H4 (H4-4Kac). In the current study, the authors extend this work to examine how H4-4Kac influences H3 N-tail dynamics in the presence of linker histone H1. Using cryo-EM and NMR, they report observations leading them to conclude that H4 N-tail acetylation modulates H1 binding, enhances interactions between the H3 N-tail and nucleosomal DNA, and consequently inhibits acetylation of the H3 N-tail.

While this topic is potentially interesting to the chromatin community, particularly researchers focused on structure–dynamics and modeling of the histone tails and their modifications, it is unclear how broadly applicable the findings are to explain existing biological observation. Additionally, the differences observed in cryo-EM and NMR are minimal. The conclusions/claims drawn from these results would need to be strengthened by additional experiments (preferably independent) and/or more rigorous analysis. The manuscript would also benefit from revisions to avoid over-interpretation of minor changes.

Major concerns:

1. When comparing the cryo-EM structures of class 1 to that of class2 (Fig 1b), the authors stated “the distal linker-DNA appears collapsed with slightly different orientations, suggesting that H4-4Kac affects the flexibility of distal linker DNA”. However, superimposition of the two maps in Fig 1b shows similar orientation rather than the “different orientation” described. Comparing such flexible and low-resolution regions in a high-resolution map directly is not appropriate. The authors should apply low-pass filtering to both maps (to ~6-7 Å) and then compare the DNA ends.
2. The observed chemical shift perturbations ($\Delta\delta$) (Fig 2d) are very small (all < 0.1 ppm, with most < 0.05 ppm). They are within a range typically associated with weak or nonspecific interactions. Based on these results, only a few amino acids in BS1 show change in chemical environment ($\Delta\delta > 0.05$ ppm) that may suggest DNA binding. I advise caution in drawing strong conclusions from such minor shifts (as done in the current version). The authors should either perform more rigorous analysis or additional experiments to support their claims. Alternatively, they can revise the conclusions to be more consistent with the results.

Minor concerns:

1. The results of the H1.4 titration experiments were mentioned but not shown. The concentration of H1.4 used for reconstituted chromatosomes was not provided in the Materials and Methods section either. This information is important for judging the proper H1-to-nucleosome ratio, which directly impact the level of linker DNA compaction in chromatosome.
2. In addition to their prior work, the authors should compare their findings with other published studies (e.g. Stützer et al., 2016) in the discussion.

Version 1:

Reviewer comments:

Reviewer #1

(Remarks to the Author)

The authors have addressed all my comments in full with analysis of cryo-EM data and Coarse-grained molecular modeling. I support publication of the revised manuscript.

Reviewer #2

(Remarks to the Author)

While the authors have walked back their interpretation slightly, they have not done so to the level needed, and are still making gross overstatements based on the data presented. For instance, in the introduction the authors still state that reference 11 shows that the tail "is switched to a more specific dynamic dyad axis, and linker- and core-DNA contact state". Again, they provided no evidence in this previous work of where the tail resides on the DNA. The statements of linker, core, or dyad specific contacts in the context of interpreting the NMR data ALL need to be removed unless the authors can specifically determine the chemical shift of different states. In addition, while the newly added CG simulations are interesting they did not experimentally validate these simulations and thus they currently are stand-alone results. The NMR data cannot be interpreted within the context of the simulations without some sort of validation. Thus, as is, this manuscript is still not suitable for publication in communications biology.

Reviewer #3

(Remarks to the Author)

The authors have added substantial new CG-MD data in the revised manuscript, greatly strengthening the paper. In combination with the other revisions, these changes satisfactorily address the concerns raised by the reviewers.

Version 2:

Reviewer comments:

Reviewer #2

(Remarks to the Author)

The authors have now sufficiently addressed all comments raised.

COMMSBIO-25-5508

Reviewers' comments:

Reviewer #1 (Remarks to the Author):

This work by Furukawa et al, using cryo-EM and NMR to study how H4 N-tail (H4-4Kac) affects H3 N-tail acetylation in the context of H1.4 chromatosome. In unmodified chromatosomes, H3 N-tails dynamically interact with both core-DNA and linker-DNA, allowing acetylation. In H4-4Kac chromatosomes, H4 tail acetylation changes the dynamics, leading to H3 N-tails predominantly interacting with core-DNA between two DNA gyres and repressing its acetylation. This work suggests that acetylation of H4 N-tail modulates H1 binding, enhancing interaction between H3 N-tail and core-DNA, and thus inhibiting H3 acetylation by the Gcn5 HAT domain. Overall, the results are clearly presented and discussed. I have a few suggestions below.

We greatly appreciate your positive comments.

1. Can the DNA base pair sequence be distinguished in the high resolution class 1 and 2 cryo-EM maps? Are class 1 and class 2 the same or not in terms of DNA sequence orientation? In the unmodified H1.4 chromatosome structure (7k5y), the DNA sequence can be assigned based on the cryo-EM density map. This point is important and could enhance the model that the author presented in Figure 3d.

Thank you for your kind comment. In response, we have refined the cryo-EM maps and revealed that the DNA sequences in class 1 and class 2 were identical orientations, as shown in new Fig. S3. We have added this point to the manuscript as indicated below.

Overall structure of the H4-4Kac chromatosome

In both structures, the proximal linker-DNA to which H1.4 binds is clearly resolved, as it is in the unmodified chromatosome. In contrast, the distal linker-DNA appears collapsed with slightly different orientations between the two classes, suggesting that H4-4Kac affects the flexibility of

the distal linker-DNA (Fig. 1B). It should be noted that the DNA sequences within these two H4-4Kac chromosome structures were determined from the cryo-EM maps, and the DNA orientation was identical in both (Fig. S3).

Fig. S3. Orientations of DNA in the two H4-4Kac chromosome structures. DNA sequences were determined from cryo-EM maps of Class 1 (left) and Class 2 (right) H4-4Kac chromosomes. Each panel shows close-up views of the cryo-EM maps around the nucleosomal dyad DNA together with the fitted model (PDB ID: 7K5Y).

2. In Figure 3a, for the NCP column, the K14ac peak doesn't show up at 10h or 20h, what causes the intensity decrease of H3K14 in NCP? I would also suggest quantifying time-dependent signal changes of H3K14ac, and list the acetylation rate constant for the exponential fittings. Are these rates relevant to in vivo histone acetylation?

Thank you for your comment. In our previous work (Ref. 14), we reported that the acetylation reaction in NCP requires a 10-fold higher enzyme concentration to observe detectable product formation. In the present study, however, we performed measurements under the same enzyme concentration as used for the nucleosome (100 nM Gcn5), and observed only a negligible rate of acetylation in the NCP. In response to your suggestion, we have provided a table of acetylation rate constants (Table 2); the quantified acetylation rates confirm that the H4-4Kac chromosome is acetylated approximately 10-fold more slowly than the nucleosome or chromosome, consistent with its NCP-like dynamics. Regarding in vivo relevance, we agree that the acetylation rates obtained under our in vitro conditions (buffer composition, NaCl concentration, temperature, and enzyme concentration) are not directly comparable to in

vivo acetylation rates. Nevertheless, the relative ordering of the rates across NCP, H4-4Kac chromosome, chromosome, and nucleosome is likely to be important in vivo conditions.

Table 2. Rate constant for acetylation of K14

	s ⁻¹
nucleosome	$4.3 \times 10^{-5} \pm 7.0 \times 10^{-8}$
chromosome	$1.7 \times 10^{-5} \pm 4.1 \times 10^{-8}$
H4-4Kac chromosome	$4.5 \times 10^{-6} \pm 5.8 \times 10^{-8}$
NCP	$6.9 \times 10^{-6} \pm 3.8 \times 10^{-7}$

3. The study indicates that it is the core domain of the linker histone H1.4 that causes the decrease in H3K14 acetylation induced by H4-4Kac, however, the exact mechanisms by which H1.4 influences the dynamics and accessibility of H3 N-tails are not clear. Understanding the specific contributions of the N- and C-terminal domains of H1.4 could further clarify the conclusion. For example, express H1.4 with N-terminal or C-terminal deletion, or both, then compare how these changes affect H3K14 tail acetylation.

Thank you for your kind comment. We fully agree with your point that the N- and C-terminal domains of H1.4 may also contribute to regulating the dynamics and accessibility of H3 N-tails. In the revised manuscript, we have reported the results of coarse-grained molecular dynamics (CG-MD) simulations to reveal the dynamic interactions of the N-and C-tails of H1.4 with the H3 N-tails and added a new Fig. S5 as indicated below.

Coarse-grained molecular models of nucleosomes and chromosomes.

In both conventional and H4-4Kac chromosomes containing H1.4, CG-MD indicated that the H3 N-tail exhibits asymmetric behavior: the distal H3 N-tail (N-Tail A) contacts only core-DNA, whereas the proximal H3 N-tail (N-Tail B) contacts both linker-DNA and core-DNA, as found in the conventional nucleosome (Fig. 2A). This asymmetry arises from the binding of the linker histone H1.4 to proximal linker-DNA, in addition to the dyad axis of the nucleosome. The N- and C-tails of H1.4 fluctuate dynamically and transiently contacting either side of the linker-DNA (Fig. S5). It is surprising that in both conventional and H4-4Kac chromosomes, the C-tail

of H1.4 contacts both core- and distal linker-DNA with similar percentages of contacts with core- and distal linker-DNA, but makes less contacts with proximal linker-DNA, while the N-tail of H1.4 primarily contacts proximal linker-DNA without any contact to core-DNA (Fig. S5E, F). This difference reflects the long C-tail relative to the short N-tail of H1.4. These stochastic fluctuations of the H1.4 tails regulate the H3 N-tail contact with DNA, hindering the contacts of the distal H3 N-tail with the linker-DNA but not that of the proximal H3 N-tail in both chromosomes.

Fig. S5. Chromatosome molecular dynamics simulations. A) Heat maps of H4 N-tail contacts with DNA in the conventional (left) and acetylated (right) chromosome. **B)** Percentage of contacts between the H4 N-tail and DNA. **C, D)** Heat maps of H1.4 linker histone with core-DNA, proximal ('p') and distal ('d') linker-DNA in the conventional (**C**) and acetylated (**D**)

chromatosome. Left: total heat maps of contacts of H1.4. Center: heat maps of contacts of the H1.4 C-tail. Right: heat maps of contacts of the H1.4 N-tail. **E, F**) Percentage of contacts between the H1.4 C-tail (left) or N-tail (right) and core-DNA or linker DNA in the conventional (**E**) and acetylated (**F**) chromatosome during 10- μ s simulations.

In support of this notion, a recent study reported that acetylation mimics of the H3 tail enhance dynamic exchange of the nucleosome-bound H1 C-terminal domain on linker-DNA, indicating that the H1 CTD can influence the dynamics and accessibility of H3 N-tails. We have therefore added the following text to the Discussion.

Discussion

In the H4-4Kac chromatosome, only NMR signals corresponding to the NCP-like H3 N-tails were observed; this suggests that H4-4Kac accelerates the dynamic exchange of H1 between the two linker-DNAs, preventing the H3 N-tails from contacting linker-DNA on the NMR timescale (Fig. 4C-D). The Cryo-EM structures of the H4-4Kac chromatosome revealed that the core configuration of the globular domain of H1.4, core-histones, and nucleosomal DNA is essentially identical to that of the unmodified chromatosome, regardless of the fluctuation timescale of the linker histone H1.4, which is on the order of either a few seconds for NMR measurements in the H4-4Kac chromatosome, or several tens of minutes for enzyme reactions in the unmodified chromatosome. In the current CG-MD simulations, both conventional and H4-4Kac chromatosomes showed that the distal H3 N-tail contacts core-DNA, while the proximal H3 N-tail contacts linker-DNA in addition to core-DNA. The simulations indicated that the distal H3 N-tail interacts with core-DNA less in the H4 4Kac chromatosome (75%) than in the conventional one (90%), and the proximal H3 N-tail contacts with linker-DNA less in the H4 4Kac chromatosome (13 %) than in the conventional one (21 %) (Fig. 2C, D and Fig. 4C). This may relate to the different binding dynamics of H1 locations in the conventional and H4-4Kac chromatosomes. In addition, it has been previously reported that acetylation mimetics of the H3 N-tail enhance dynamic exchange of the nucleosome-bound H1 C-tail on linker-DNA, supporting the notion that the H1 C-tail can modulate both the dynamics and accessibility of H3 N-tails (22).

References

22. S. K. Das, *et al.*, Histone H3 Tail Modifications Alter Structure and Dynamics of the H1 C-Terminal Domain Within Nucleosomes. *Journal of Molecular Biology* **435**, 168242 (2023).

Minor comments:

Figure 3a, H4ac_chromatosome column, should the NMR peak be shown in black color for consistency?

Thank you for this comment; however, in old Figure 3A (new Figure 4A), the H4ac_chromatosome spectra are already shown in black.

Figure 34. Effect of H4-4Kac on the H3 N-tail acetylation rate and model of H3 N-tail–DNA interactions in the H4-4Kac chromatosome. A) Time-dependent signal changes of K14 and acetylated K14 of the H3 N-tail in the nucleosome (red), chromatosome (green), H4-4Kac chromatosome (black), and NCP (blue) after the addition of Gcn5.

Reviewer #2 (Remarks to the Author):

In this manuscript cryoEM and NMR spectroscopy are utilized to interrogate the effects of the H1.4-chromatosome structure and H4 tail acetylation on the acetylation of the H3 tail by Gcn5. The study is timely as there is a growing

interest in how the nucleosome regulates histone tail accessibility and how variations within further alter this. Overall, the results are sound and informative with respect to cross talk between H1.4 and the H3 tail as well as between the H3 and H4 tails. However, the authors overinterpret their NMR data with respect to molecular models of the tails themselves. In particular, there are a number of statements regarding the residence of the H3 and H4 tails on the double-helical gyres of the nucleosome core, the dyad, or the linker DNA that are not supported by data. The chemical shifts observed cannot be assigned to a particular structural model without more data. Though the authors can assign states (e.g. NCP-state, nucleosome-state, etc.) they cannot interpret at the molecular level where the tail is residing without additional data. A revised manuscript in which these statements are walked back would be of great interest to the community. Below are examples of statements that need to be rephrased. This list is representative but not inclusive and the full text should be revisited in this light. In addition, I have one question regarding Gcn5 activity.

Thank you for your critical comment. To support our interpretation of the H3 N-tail contact states, we have performed coarse-grained molecular dynamics (CG-MD) simulations of conventional as well as H4-4Kac chromatosomes and nucleosomes. The heat contact maps of the proximal and distal H3 N-tails in both chromatosomes have revealed asymmetrical core- and linker-DNA contact states as shown in new Fig.2.

Coarse-grained molecular models of nucleosomes and chromatosomes.

To reveal the dynamics of the H3 N-tails, we performed CG-MD simulations on the conventional and H4-4Kac nucleosomes, as well as on the conventional and H4-4Kac chromatosomes with H1.4. The simulations showed that, in the conventional nucleosome, both H3 N-tails interact with linker-DNA in addition to core-DNA (Fig. 2A), indicating that the N-tail is in the LC state, as suggested by earlier NMR experiments. On H4-4Kac, the H4 N-tails are released from their contact with core-DNA (Fig. S4C), and the contact maps of the H3 N-tail reveal subtle differences, albeit not significant, in several regions (Fig. 2B and Fig. S4B). These observations seem to correspond to our previous NMR experiments (11).

Figure 2. CG-MD simulation of H3 N-tail interactions with DNA. A) Heat contact maps of the two H3 N-tails in the conventional nucleosome, showing side and top views of the nucleosome (left) and the percentage of contacts between the H3 N-tail and core- and linker-DNA (right). **B)** Difference contact maps of the H3 N-tails in the conventional nucleosome and those in the H4-4Kac nucleosomes, with positive values indicated in blue and negative ones in red. **C, D)** Heat contact maps of the proximal H3 N-tail (N-Tail B) and the distal H3 N-tail (N-Tail A) with core-DNA, proximal linker-DNA ('p'), and distal linker-DNA ('d') in the conventional (**C**) and H4-4Kac

(D) chromosomes, showing top and side views of the chromosome structures (left) and the percentages of contacts during 10 μ s simulations (right).

In both conventional and H4-4Kac chromosomes containing H1.4, CG-MD indicated that the H3 N-tail exhibits asymmetric behavior: the distal H3 N-tail (N-Tail A) contacts only core-DNA, whereas the proximal H3 N-tail (N-Tail B) contacts both linker-DNA and core-DNA, as found in the conventional nucleosome (Fig. 2A). This asymmetry arises from the binding of the linker histone H1.4 to proximal linker-DNA, in addition to the dyad axis of the nucleosome. The N- and C-tails of H1.4 fluctuate dynamically and transiently contacting either side of the linker-DNA (Fig. S5). It is surprising that in both conventional and H4-4Kac chromosomes, the C-tail of H1.4 contacts both core- and distal linker-DNA with similar percentages of contacts with core- and distal linker-DNA, but makes less contacts with proximal linker-DNA, while the N-tail of H1.4 primarily contacts proximal linker-DNA without any contact to core-DNA (Fig. S5E, F). This difference reflects the long C-tail relative to the short N-tail of H1.4. These stochastic fluctuations of the H1.4 tails regulate the H3 N-tail contact with DNA, hindering the contacts of the distal H3 N-tail with the linker-DNA but not that of the proximal H3 N-tail in both chromosomes.

Although CG-MD simulations indicated that the dynamics of the H3 N-tails are very similar between the conventional and H4-4Kac chromosomes within the 10- μ s simulations applied, there are some differences: the percentages of contacts of the distal H3 N-tail (N-tail A) with core-DNA is ~90 % in the conventional chromosome and ~75 % in the acetylated one; in addition, the percentages of the proximal H3 N-tail (N-tail B) contacts with core- and linker-DNA is, respectively, ~64 % and ~21 %, in the conventional chromosome, and ~57 % and ~13 % in the acetylated one (Fig. 2C, D).

The CG-MD simulations revealed that the H4 N-tail with 4Kac is released from core-DNA into a solvent-exposed state (Fig. S5A, B); this modification influences the dynamics of the H1.4 tails, thereby resulting in the slightly different H3 N-tail behavior found in both systems as suggested by previous NMR experiments.

1) The statement “Specifically, the dynamic linker–DNA contact state (hereafter “L–state”) is switched to a more specific dynamic dyad axis DNA and linker–DNA contact state (hereafter “Ld–state”), which is related to the formation of euchromatin (11).” is not supported by the publication referenced. While the authors previously showed changes to the conformational ensemble of the H3 tail upon acetylation of the H4 tail there was no data to support specific residence time on either linker or dyad. The authors should not rephrase their previous findings in

this way without a thorough reanalysis and/or additional data to support their statements.

Thank you for your critical comment. In response, in our revised manuscript we have deleted “Ld state”. In addition, we have performed CG-MD simulations.

Introduction

Previously, using NMR, we examined the dynamics of the H3 N-tails in nucleosomes with and without tetra-acetylation of the H4 N-tail at K5, K8, K12 and K16 (H4-4Kac) and compared the rate of H3 K14 acetylation by the histone acetyltransferase (HAT) domain of Gcn5. We found that H4-4Kac alters the dynamic state of the N3 N-tail on linker-DNA, enhancing its acetylation at K14 by Gcn5. Specifically, the dynamic linker- **and core**-DNA contact state (hereafter “LC-state”) is switched to a more specific dynamic dyad axis DNA, and linker- **and core**-DNA contact state (hereafter “Ld-state”), which is related to the formation of euchromatin (11).

Coarse-grained molecular models of nucleosomes and chromatosomes.

To reveal the dynamics of the H3 N-tails, we performed CG-MD simulations on the conventional and H4-4Kac nucleosomes, as well as on the conventional and H4-4Kac chromatosomes with H1.4. The simulations showed that, in the conventional nucleosome, both H3 N-tails interact with linker-DNA in addition to core-DNA (Fig. 2A), indicating that the N-tail is in the LC state, as suggested by earlier NMR experiments. On H4-4Kac, the H4 N-tails are released from their contact with core-DNA (Fig. S4C), and the contact maps of the H3 N-tail reveal subtle differences, albeit not significant, in several regions (Fig. 2B and Fig. S4B). These observations seem to correspond to our previous NMR experiments (11).

2) It is unclear how a “shift toward to the corresponding signals of the NCP” indicates “increased DNA contact”. The authors provide no explanation or evidence for this.

Thank you for your critical comment. In the revised manuscript, we simplified this description and noted that these residues shift toward the corresponding signals of the NCP.

H4-4Kac affects the dynamics of the histone H3 N-tails in the chromatosome.

Here, differences in chemical shift between the H4-4Kac nucleosome and NCP were relatively small in BS1, L1, BS2₁, and BS2₂. In the unmodified nucleosome, by contrast, residues in BS2₂ and L2 together with K18 and Q19 in the H3 N-tails shift toward to the corresponding signals of the NCP (14), ~~suggesting that BS2₂ shows increased DNA contact in the nucleosome.~~

3) The authors note “Previous studies have shown that the H3 N-tails in NCP make dynamic and robust contact with two gyres of core-DNA”. However, these studies also show that the H3 tails can contact the dyad in the nucleosomal context. Thus interpretation of the H3 N-tails being entirely two gyre bound in the NCP does not appear warranted.

Thank you for your critical comment. We agree that our original wording was too strong. In response, we have deleted “the two gyres of core DNA in the NCP” in the revised manuscript.

Rate of histone H3 N-tail acetylation in the H4-4Kac nucleosome.

Previous studies have shown that the H3 N-tails in NCP make dynamic and robust contact with ~~two gyres of core-DNA~~ (10,12-15), adopting the C-state. In this state, acetylation of H3 K14 by the Gcn5 HAT domain is greatly suppressed relative to that in the nucleosome, where the H3 N-tails additionally contact the linker-DNA (9,14,15) in the LC-state, which is more accessible to the enzyme (14).

Discussion

Here, in the case of the H4-4Kac nucleosome, two distinct H3 N-tail NMR signals were observed for the BS2₂ region, along with an increased rate of H3 K14 acetylation relative to the unmodified nucleosome. We previously proposed that two different linker-DNA contact states exist in the H4-4Kac nucleosome (11,14), ~~termed the L-state and Ld-state.~~ **One** The L-state corresponds to the unmodified nucleosome-like state (**LC-state**), with the H3 N-tail protruding dynamically ~~between two gyres of~~ **contacting** core-DNA and additionally ~~contacting~~ linker-DNA. In the **other** Ld-state, the BS2₂ region of the H3 N-tail dynamically contacts ~~the region near the dyad axis of core-DNA,~~ which was previously and dynamically occupied by the unmodified H4 N-tail within the above-mentioned fuzzy complex. ~~In addition, the BS1 region of the H3 N-tail before the BS2 region dynamically and additionally contacts linker DNA (Fig. 3c), resulting in~~

~~enhanced acetylation of H3 K14 in the H4-4Kac nucleosome (11,14). Our CG-MD simulations showed small but subtle difference in the H3 N-tail contacts with core- and linker-DNA between the conventional and H4-4Kac nucleosomes, which might be related to the different NMR signals (Fig. 3B and Fig. S4B) (11).~~

4) The authors propose in the discussion that the H3 tail in the H4-4Kac nucleosome will contact the dyad as the H4 tail acetylation has displaced it from the dyad. However, I am unclear where the concept that the H4 tail is at the dyad comes from. Please cite the literature or data that indicates that the unmodified H4 tail can contact the dyad.

Thank you for helpful comments. In response, we have performed CG-MD simulations of the H4 N-tail and revised the previous description that “the unmodified H4 tail can occupy the dyad region”, as indicated below.

Coarse-grained molecular models of nucleosomes and chromosomes.

To reveal the dynamics of the H3 N-tails, we performed CG-MD simulations on the conventional and H4-4Kac nucleosomes, as well as on the conventional and H4-4Kac chromosomes with H1.4. The simulations showed that, in the conventional nucleosome, both H3 N-tails interact with linker-DNA in addition to core-DNA (Fig. 2A), indicating that the N-tail is in the LC state, as suggested by earlier NMR experiments. On H4-4Kac, the H4 N-tails are released from their contact with core-DNA (Fig. S4C), and the contact maps of the H3 N-tail reveal subtle differences, albeit not significant, in several regions (Fig. 2B and Fig. S4B). These observations seem to correspond to our previous NMR experiments (11).

The CG-MD simulations revealed that the H4 N-tail with 4Kac is released from core-DNA into a solvent-exposed state (Fig. S5A, B); this modification influences the dynamics of the H1.4 tails, thereby resulting in the slightly different H3 N-tail behavior found in both systems as suggested by previous NMR experiments.

5) The authors should comment on how the presence of linker DNA or the H4 tail may inherently effect Gcn5 activity outside of H3 tail accessibility.

Thank you for your critical comment. In our study, we focused on reduced H3-tail accessibility as the most direct explanation for the decreased efficiency of Gcn5-mediated acetylation observed upon H4 acetylation. We

acknowledge that H4 acetylation or the presence of linker-DNA might in principle also affect Gcn5 activity through other factors, such as changes in the electrostatic environment or nucleosome architecture. While our current data do not allow us to separate these contributions, we agree that they remain possible and should be considered in future studies.

Reviewer #3 (Remarks to the Author):

The manuscript builds on two previous studies by the same research group, where NMR was used to characterize the conformations of the histone H3 N-terminal tail (H3 N-tail) and its interplay with tetra-acetylation of histone H4 (H4-4Kac). In the current study, the authors extend this work to examine how H4-4Kac influences H3 N-tail dynamics in the presence of linker histone H1. Using cryo-EM and NMR, they report observations leading them to conclude that H4 N-tail acetylation modulates H1 binding, enhances interactions between the H3 N-tail and nucleosomal DNA, and consequently inhibits acetylation of the H3 N-tail. While this topic is potentially interesting to the chromatin community, particularly researchers focused on structure–dynamics and modeling of the histone tails and their modifications, it is unclear how broadly applicable the findings are to explain existing biological observation. Additionally, the differences observed in cryo-EM and NMR are minimal. The conclusions/claims drawn from these results would need to be strengthened by additional experiments (preferably independent) and/or more rigorous analysis. The manuscript would also benefit from revisions to avoid over-interpretation of minor changes.

Thank you for your critical comment. To support our interpretation of the H3 N-tail contact states, we have performed coarse-grained molecular dynamics (CG-MD) simulations in conventional as well as H4-4Kac chromosomes and nucleosomes.

Major concerns:

1. When comparing the cryo-EM structures of class 1 to that of class2 (Fig 1b), the authors stated “the distal linker-DNA appears collapsed with slightly different orientations, suggesting that H4-4Kac affects the flexibility of distal linker DNA”. However, superimposition of the two maps in Fig 1b shows similar orientation

rather than the “different orientation” described. Comparing such flexible and low-resolution regions in a high-resolution map directly is not appropriate. The authors should apply low-pass filtering to both maps (to ~6-7 Å) and then compare the DNA ends.

Thank you for your critical comment. In response, we have applied low-pass filtering as indicated below.

Figure 1. Cryo-EM structures of H4-4Kac chromosome. A) Class 1 (i) and class 2 (ii) cryo-EM structures of the H4-4Kac chromosome. Close-up views of the cryo-EM density maps fitted to the structural model (PDB ID: 7K5Y) are shown above. **B)** Superimposition of the two classes of cryo-EM structures of the H4-4Kac chromosome. **Both maps were low-pass-filtered to a resolution of 6.0 Å.**

2. The observed chemical shift perturbations ($\Delta\delta$) (Fig 2d) are very small (all < 0.1 ppm, with most < 0.05 ppm). They are within a range typically associated with weak or nonspecific interactions. Based on these results, only a few amino acids

in BS1 show change in chemical environment ($\Delta\delta > 0.05$ ppm) that may suggest DNA binding. I advise caution in drawing strong conclusions from such minor shifts (as done in the current version). The authors should either perform more rigorous analysis or additional experiments to support their claims. Alternatively, they can revise the conclusions to be more consistent with the results.

Thank you for your critical comment. We agree that the observed chemical shift perturbation ($\Delta\delta$) values are small (mostly <0.05 ppm) and thus should not be overinterpreted as direct evidence of strong or specific DNA binding. We also consider this interaction to be weak. We view the observed $\Delta\delta$ values as subtle but reproducible indicators of local environmental changes in the intrinsically disordered H3 tails, which is consistent with previous studies where $\Delta\delta$ values of <0.1 ppm were commonly observed for histone tails. In addition, we have performed additional CG-MD simulations to reveal the dynamics of H3 N-tails.

Coarse-grained molecular models of nucleosomes and chromatosomes.

To reveal the dynamics of the H3 N-tails, we performed CG-MD simulations on the conventional and H4-4Kac nucleosomes, as well as on the conventional and H4-4Kac chromatosomes with H1.4. The simulations showed that, in the conventional nucleosome, both H3 N-tails interact with linker-DNA in addition to core-DNA (Fig. 2A), indicating that the N-tail is in the LC state, as suggested by earlier NMR experiments. On H4-4Kac, the H4 N-tails are released from their contact with core-DNA (Fig. S4C), and the contact maps of the H3 N-tail reveal subtle differences, albeit not significant, in several regions (Fig. 2B and Fig. S4B). These observations seem to correspond to our previous NMR experiments (11).

Figure 2. CG-MD simulation of H3 N-tail interactions with DNA. A) Heat contact maps of the two H3 N-tails in the conventional nucleosome, showing side and top views of the nucleosome (left) and the percentage of contacts between the H3 N-tail and core- and linker-DNA (right). **B)** Difference contact maps of the H3 N-tails in the conventional nucleosome and those in the H4-4Kac nucleosomes, with positive values indicated in blue and negative ones in red. **C, D)** Heat contact maps of the proximal H3 N-tail (N-Tail B) and the distal H3 N-tail (N-Tail A) with core-DNA, proximal linker-DNA ('p'), and distal linker-DNA ('d') in the conventional (**C**) and H4-4Kac

(D) chromosomes, showing top and side views of the chromosome structures (left) and the percentages of contacts during 10 μ s simulations (right).

In both conventional and H4-4Kac chromosomes containing H1.4, CG-MD indicated that the H3 N-tail exhibits asymmetric behavior: the distal H3 N-tail (N-Tail A) contacts only core-DNA, whereas the proximal H3 N-tail (N-Tail B) contacts both linker-DNA and core-DNA, as found in the conventional nucleosome (Fig. 2A). This asymmetry arises from the binding of the linker histone H1.4 to proximal linker-DNA, in addition to the dyad axis of the nucleosome. The N- and C-tails of H1.4 fluctuate dynamically and transiently contacting either side of the linker-DNA (Fig. S5). It is surprising that in both conventional and H4-4Kac chromosomes, the C-tail of H1.4 contacts both core- and distal linker-DNA with similar percentages of contacts with core- and distal linker-DNA, but makes less contacts with proximal linker-DNA, while the N-tail of H1.4 primarily contacts proximal linker-DNA without any contact to core-DNA (Fig. S5E, F). This difference reflects the long C-tail relative to the short N-tail of H1.4. These stochastic fluctuations of the H1.4 tails regulate the H3 N-tail contact with DNA, hindering the contacts of the distal H3 N-tail with the linker-DNA but not that of the proximal H3 N-tail in both chromosomes.

Although CG-MD simulations indicated that the dynamics of the H3 N-tails are very similar between the conventional and H4-4Kac chromosomes within the 10- μ s simulations applied, there are some differences: the percentages of contacts of the distal H3 N-tail (N-tail A) with core-DNA is ~90 % in the conventional chromosome and ~75 % in the acetylated one; in addition, the percentages of the proximal H3 N-tail (N-tail B) contacts with core- and linker-DNA is, respectively, ~64 % and ~21 %, in the conventional chromosome, and ~57 % and ~13 % in the acetylated one (Fig. 2C, D).

The CG-MD simulations revealed that the H4 N-tail with 4Kac is released from core-DNA into a solvent-exposed state (Fig. S5A, B); this modification influences the dynamics of the H1.4 tails, thereby resulting in the slightly different H3 N-tail behavior found in both systems as suggested by previous NMR experiments.

Minor concerns:

1. The results of the H1.4 titration experiments were mentioned but not shown. The concentration of H1.4 used for reconstituted chromosomes was not provided in the Materials and Methods section either. This information is important for judging the proper H1-to-nucleosome ratio, which directly impact the level of linker DNA compaction in chromosome.

In the revised manuscript, we have now stated the concentration of H1.4 used for chromosome reconstitution in the Materials and Methods section. In addition, we have included the results of a DNA electrophoresis mobility shift assay in the Supporting Information (Fig. S6). These results confirm that, under our experimental conditions, nucleosomes are fully bound by H1.4, with no detectable free nucleosomes remaining.

NMR experiments

NMR experiments were performed on AVANCE 600-MHz and AVANCE III HD 950-MHz spectrometers with a triple-resonance TCI cryogenic probe (Bruker Bio Spin) at 298 K using 20-80 μM samples dissolved in 25 mM MES (pH 6.0), 1 mM DTT, and 5% D_2O . The unlabeled linker histone H1.4 was added step-by-step to reach ratios of 0.15, 0.3, 0.5, 1.0, and 1.5 to the 40 μM nucleosome containing $^{15}\text{N}/^2\text{H}$ -labeled H3. In each step, any precipitate that formed during the titration process was removed by centrifugation and the ^1H - ^{15}N HSQC spectrum of the supernatant of the nucleosome/H1.4 solution was acquired at 298K. In the ^1H - ^{15}N HSQC spectrum with 1.5 equivalents of H1.4 (60 μM) added, we confirmed that there were no nucleosome signals, indicating all the nucleosomes in the NMR solution were in the H1-bound form (i.e., chromosome). In addition, we performed electrophoretic mobility shift assays of H1.4 binding to each of the conventional and H4 K4ac nucleosomes to monitor the corresponding chromosome formation (Fig. S6).

Fig. S6. DNA electrophoresis mobility shift assay of linker histone H1.4 binding to the nucleosome or H4-4Kac nucleosome. Nucleosomes with linker histone H1.4 added in ratios of 0.5, 1.0, 1.5, and 2.0 were subjected to electrophoresis at 4 °C on a 7.5% native-PAGE in 1 × Tris-glycine buffer and were visualized by SYBR Gold nucleic acid gel stain.

2. In addition to their prior work, the authors should compare their findings with other published studies (e.g. Stützer et al., 2016) in the discussion.

Thank you for helpful comment. In the revised manuscript, we have mentioned these studies as indicated below.

Discussion

In the unmodified chromosome, the core domain of the linker histone H1.4 is located at the dyad axis of core-DNA (6), and the N-tail of H1.4 is thought to bind to the proximal linker-DNA. A previous study of the H3 N-tail as a free peptide, and in the nucleosome and chromosome showed that the H3 N-tail transiently and electrostatically contacts nucleosome DNA, and its dynamics are reduced by the binding of linker histone H1, depending on the C-terminal domain of H1 (9). Interestingly, our CG-MD simulations showed that the H1 C-tail contacts core-DNA in addition to the distal linker-DNA, while the H1 N-tail contacts the proximal linker-DNA and has less contact with core-DNA (Fig. S5).

Reviewers' comments:

Reviewer #1 (Remarks to the Author):

The authors have addressed all my comments in full with analysis of cryo-EM data and Coarse-grained molecular modeling. I support publication of the revised manuscript.

We greatly appreciate your positive comments.

Reviewer #2 (Remarks to the Author):

While the authors have walked back their interpretation slightly, they have not done so to the level needed, and are still making gross overstatements based on the data presented. For instance, in the introduction the authors still state that reference 11 shows that the tail "is switched to a more specific dynamic dyad axis, and linker- and core-DNA contact state". Again, they provided no evidence in this previous work of where the tail resides on the DNA. The statements of linker, core, or dyad specific contacts in the context of interpreting the NMR data ALL need to be removed unless the authors can specifically determine the chemical shift of different states. In addition, while the newly added CG simulations are interesting they did not experimentally validate these simulations and thus they currently are stand-alone results. The NMR data cannot be interpreted within the context of the simulations without some sort of validation. Thus, as is, this manuscript is still not suitable for publication in communications biology.

Thank you for your critical comment. In response, in our revised manuscript we have deleted the sentence of "the tail is switched to a more specific dynamic dyad axis, and linker- and core-DNA contact state". Furthermore, we have deleted all descriptions on "C-state" and "LC-state".

Introduction

Previously, using NMR, we examined the dynamics of the H3 N-tails in nucleosomes with and without tetra-acetylation of the H4 N-tail at K5, K8, K12 and K16 (H4-4Kac) and compared the rate of the H3 K14 acetylation by the histone acetyltransferase (HAT) domain of Gcn5. We found that H4-4Kac alters the dynamic state of the N3 N-tail on linker-DNA, enhancing its acetylation at K14 by Gcn5. Specifically, the dynamic linker- and core-DNA contact state (hereafter “LC-state”) is switched to a more specific dynamic dyad axis DNA, and linker- and core-DNA contact state (11). In the unmodified chromosome, by contrast, linker histone H1.4 induces asymmetric dynamic states of the H3 N-tail, whereby one H3 N-tail dynamically and robustly contacts core-DNA, resulting in a less accessible conformation to the enzyme (hereafter “C-state”), while the other dynamically contacts linker-DNA in addition to core-DNA, resulting in the enzyme-accessible LC-state. However, the **asymmetric** dynamic C- and LC-states of the H3 N-tails equilibrate on the time scale of the enzyme reaction, with the result that the acetylation rate of H3 K14 by Gcn5 in the unmodified chromosome is comparable to that in the nucleosome.

Coarse-grained molecular models of nucleosomes and chromosomes.

To reveal the dynamics of the H3 N-tails, we performed CG-MD simulations on the conventional and H4-4Kac nucleosomes, as well as on the conventional and H4-4Kac chromosomes with H1.4. The simulations showed that, in the conventional nucleosome, both H3 N-tails interact with linker-DNA in addition to core-DNA (Fig. 2A), indicating that the N-tail is in the LC-state, as suggested by earlier NMR experiments.

Discussion

Here, in the case of the H4-4Kac nucleosome, two distinct H3 N-tail NMR signals were observed for the BS2₂ region, along with an increased rate of H3 K14 acetylation relative to the unmodified nucleosome. We previously proposed that two different linker-DNA contact states exist in the H4-4Kac nucleosome (11,14). One state corresponds to the unmodified-nucleosome-like state (LC-state), with the H3 N-tail dynamically contacting core-DNA and additionally linker-DNA. In the other state, the BS2₂ region of the H3 N-tail dynamically contacts DNA, which was previously and dynamically occupied by the unmodified H4 N-tail within the above-mentioned fuzzy complex. Our CG-MD simulations showed small but subtle difference in the H3 N-tail contacts with core- and linker-DNA between the conventional and H4-4Kac nucleosomes, which might be related to the different NMR signals (Fig. 3B and Fig. S4B) (11).

In addition, we have addressed previous NMR results, supported by other experiments and MD simulations, in “Introduction” by citing additional references.

Introduction

In most cases, however, the structures of the histone tails remain unresolved, although their dynamics and PTMs have been studied by nuclear magnetic resonance (NMR) spectroscopy (7-2147).

Notably, the two-dimensional ^1H and ^{15}N NMR signals of the H3 N-tail in the nucleosome were closer to those of an H3 N-tail peptide (residues 1–33) fused to the globular protein GB1 in its DNA-bound form than to those of the peptide in its DNA-free form (9), indicating that the H3 N-tail in the nucleosome transiently binds to linker DNA. This conclusion was further supported by H3 N-tail modification experiments. Even in a nucleosome lacking linker DNA, i.e., in a nucleosome core particle (NCP), the H3 N-tail signals more closely resembled those of an H3 N-tail peptide (residues 1–44) bound to an H3-tailless NCP than those of the free peptide (10). Consistent with these experimental data, MD simulations demonstrated that the H3 N-tail robustly and dynamically packs onto the core DNA (10). In addition, interactions of both the H4 and H3 N-tails with nucleosome DNA have been reported to regulate histone-tail modifications (21). At high salt concentrations, which weaken electrostatic interactions between the H3 N-tail and DNA, nearly all H3 N-tail signals in both the NCP and nucleosome shifted toward those of the DNA-free H3 N-tail fragment peptide (15). These interactions are also consistent with a previous integrative study combining NMR, chemical reactivity, MD, and fluorescence analyses (19), as well as single-molecule FRET experiments (22) and MD simulations (23-28). Interestingly, however, almost all H3 N-tail signals in the NCP exhibited a slight but significant high-field shift relative to the corresponding signals in the nucleosome, moving even further away from the free-peptide signals in the ^1H and/or ^{15}N dimensions (10, 13, 14). These observations indicate that the H3 N-tail dynamically contacts DNA with distinct interaction characteristics in the NCP and nucleosome contexts. These characteristics correlate with the markedly different rate constants of H3K14 acetylation by the histone acetyltransferase (HAT) domain of Gcn5: in the NCP, the acetylation rate of H3K14 is greatly reduced compared with that in the nucleosome (11, 14).

To support our CG-MD simulations, we have also cited previous findings from several experiments and MD simulations.

Coarse-grained molecular models of nucleosomes and chromatosomes.

To reveal the dynamics of the H3 N-tails, we performed CG-MD simulations on the conventional and H4-4Kac nucleosomes, as well as on the conventional and H4-4Kac chromatosomes with H1.4. The simulations showed that, in the conventional nucleosome, both H3 N-tails interact with linker-DNA in addition to core-DNA (Fig. 2A), ~~indicating that the N-tail is in the LC state~~, as suggested by earlier NMR experiments. **In addition, H4 N-tail extensively contacts DNA, consistent with previous NMR, cross-linking, and MD studies (16-18, 21, 29-31).** On H4-4Kac, the H4 N-tails are released from their contact with core-DNA (Fig. S4C), **in agreement with recent electrostatic potential analyses based on NMR data (20) and with our previous NMR results showing that acetylation increases the signal intensities of H4 N-tails, indicative of enhanced tail flexibility (14).** ~~and the~~ The contact maps of the H3 N-tail reveal subtle differences, albeit not significant, in several regions (Fig. 2B and Fig. S4B). These observations seem to correspond to our previous NMR experiments (11).

In both conventional and H4-4Kac chromatosomes containing H1.4, CG-MD indicated that the H3 N-tail exhibits asymmetric behavior: the distal H3 N-tail (N-Tail A) contacts only core-DNA, whereas the proximal H3 N-tail (N-Tail B) contacts both linker-DNA and core-DNA, as found in the conventional nucleosome (Fig. 2A). This asymmetry arises from the binding of the linker histone H1.4 to proximal linker-DNA, in addition to the dyad axis of the nucleosome. The N- and C-tails of H1.4 fluctuate dynamically and transiently contacting either side of the linker-DNA (Fig. S5). ~~It is surprising that in~~ **In both conventional and H4-4Kac chromatosomes, the C-tail of H1.4 contacts both core- and distal linker-DNA with similar percentages of contacts with core- and distal linker-DNA, but makes less contacts with proximal linker-DNA, while the N-tail of H1.4 primarily contacts proximal linker-DNA without any contact to core-DNA (Fig. S5E, F).** This difference reflects the long C-tail relative to the short N-tail of H1.4. **Consistently, in conventional chromatosomes, previous cryo-EM structures showed that the H1 C-tail binds primarily to distal linker DNA (5), and earlier CG-MD simulations demonstrated that the H1 C-tail dynamically associates with distal linker DNA rather than proximal linker DNA, in addition to dyad DNA (32, 33).** These stochastic fluctuations of the H1.4 tails regulate the H3 N-tail contact with DNA, hindering the contacts of the distal H3 N-tail with the linker-DNA but not that of the proximal H3 N-tail in both chromatosomes. **The asymmetric structures of the H3 N-tails are consistent with recent MD simulations reporting distinct DNA-interaction modes of the H3 N-tails in the chromatosome (28).** Furthermore, acetylation mimetics of the H3 N-tail were reported to enhance the dynamic exchange of the nucleosome-bound H1 C-tail on linker DNA, supporting the idea that the H1 C-tail modulates both the dynamics and accessibility of the H3 N-tails (34).

References

-
18. Bolik-Coulon, N., Rößler, P. & Kay, L. E. NMR-Based Measurements of Site-Specific Electrostatic Potentials of Histone Tails in Nucleosome Core Particles. *J. Am. Chem. Soc.* **147**, 14519–14529 (2025).
 19. Gatchalian, J. et al. Accessibility of the histone H3 tail in the nucleosome for binding of paired readers. *Nat Commun* **8**, 1489 (2017).
 20. Bolik-Coulon, N., Rößler, P., Nosella, M. L., Kim, T. H. & Kay, L. E. Modulation of histone tail electrostatic potentials in nucleosome core particles by acetylation and PARylation. *Proc. Natl. Acad. Sci. U.S.A.* **122**, e2511507123 (2025).
 21. Marunde, M. R. et al. Nucleosome conformation dictates the histone code. *eLife* **13**, e78866 (2024).
 22. Lehmann, K. et al. Dynamics of the nucleosomal histone H3 N-terminal tail revealed by high precision single-molecule FRET. *Nucleic Acids Research* **48**, 1551–1571 (2020).
 23. Peng, Y., Li, S., Onufriev, A., Landsman, D. & Panchenko, A. R. Binding of regulatory proteins to nucleosomes is modulated by dynamic histone tails. *Nat Commun* **12**, 5280 (2021).
 24. Ikebe, J., Sakuraba, S. & Kono, H. H3 Histone Tail Conformation within the Nucleosome and the Impact of K14 Acetylation Studied Using Enhanced Sampling Simulation. *PLoS Comput Biol* **12**, e1004788 (2016).
 25. Shaytan, A. K. et al. Coupling between Histone Conformations and DNA Geometry in Nucleosomes on a Microsecond Timescale: Atomistic Insights into Nucleosome Functions. *Journal of Molecular Biology* **428**, 221–237 (2016).
 26. Li, Z. & Kono, H. Distinct Roles of Histone H3 and H2A Tails in Nucleosome Stability. *Sci Rep* **6**, 31437 (2016).
 27. Zandian, M. et al. Conformational Dynamics of Histone H3 Tails in Chromatin. *J. Phys. Chem. Lett.* **12**, 6174–6181 (2021).
 28. Portillo-Ledesma, S., Li, Z. & Schlick, T. Molecular dynamics simulations reveal subtle consequences of H3K9 and H3K27 tri-methylation on chromatin constituents. *Biophysical Journal* **124**, 4005-4017 (2025).
 29. Murphy, K. J. et al. HMGN1 and 2 remodel core and linker histone tail domains within chromatin. *Nucleic Acids Research* **45**, 9917–9930 (2017).
 30. Karch, K. R. et al. Hydrogen-Deuterium Exchange Coupled to Top- and Middle-Down Mass Spectrometry Reveals Histone Tail Dynamics before and after Nucleosome Assembly. *Structure* **26**, 1651-1663.e3 (2018).
 31. Mullahoo, J. et al. Dual protease type XIII/pepsin digestion offers superior resolution and overlap for the analysis of histone tails by HX-MS. *Methods* **184**, 135–140 (2020).

32. ~~19~~-Wu, H., Dalal, Y., and Papoian, G. Binding Dynamics of Disordered Linker Histone H1 with a Nucleosomal Particle. *J Mol Biol* **433**, 166881 (2021).
33. Sridhar, A. et al. Emergence of chromatin hierarchical loops from protein disorder and nucleosome asymmetry. *Proc. Natl. Acad. Sci. U.S.A.* **117**, 7216–7224 (2020).
34. ~~22~~-Das, S. K. et al., Histone H3 Tail Modifications Alter Structure and Dynamics of the H1 C-Terminal Domain Within Nucleosomes. *J. Mol. Biol.* **435**, 168242 (2023).
35. ~~18~~-Woods, D., and Wereszczynski, J. Elucidating the influence of linker histone variants on chromosome dynamics and energetics. *Nucleic Acids Res* **48**, 3591-3604 (2020).
36. ~~20~~-Zhang, H., Huo, Q., Gao, Y. DNA Sequence-Dependent Binding of Linker Histone gH1 Regulates Nucleosome Conformations. *J. Phys. Chem. B* **126**, 6771–6779 (2022).
37. ~~21~~-Misteli, T., Gunjan, A., Hock, R., Bustin, M., and Brown, D. Dynamic binding of histone H1 to chromatin in living cells. *Nature* **408**, 877-881 (2000).
38. ~~23~~-Perrella G., et al., The Histone Deacetylase Complex 1 Protein of Arabidopsis Has the Capacity to Interact with Multiple Proteins Including Histone 3-Binding Proteins and Histone Variants. *Plant Physiol* **171**, 62-70 (2016).

Reviewer #3 (Remarks to the Author):

The authors have added substantial new CG-MD data in the revised manuscript, greatly strengthening the paper. In combination with the other revisions, these changes satisfactorily address the concerns raised by the reviewers.

We greatly appreciate your positive comments.